# Universal efficiency boost in prethermal quantum heat engines at negative temperature

Alberto Brollo [1,2], Adolfo del Campo[3,4] & Alvise Bastianello [2,5] ✉

Heat engines near the adiabatic limit typically assume a working medium at thermal equilibrium. However, quantum many-body systems often showcase conservation laws that hinder thermalization, leading to prethermalization in exotic stationary phases. This work explores whether prethermalization enhances or reduces engine efficiency. We investigate Otto cycles in quantum systems with varying numbers of conserved quantities. We find that additional conservation laws reduce efficiency at positive temperatures, but enhance it in regimes of negative temperatures. Our findings stem from general thermodynamic inequalities for infinitesimal cycles, and we provide evidence for integrable models undergoing finite cycles using the theoretical framework of Generalized Hydrodynamics. The relevance of our results for quantum simulators is also discussed.

The analysis of heat engines has played a key role since the birth of thermodynamics[1]. The advent of quantum thermodynamics has followed a similar path, with the design and characterization of quantum heat engines[2,3]. Early theoretical proposals[4] have been adapted for their implementation with current platforms for quantum technologies, including trapped ions[5–7], nitrogen-vacancy centers[8], ultracold gases[9], and NMR systems[10]. Technological advances have motivated studies beyond canonical equilibrium involving coherence, squeezing, negative temperatures[11–15], and genuine nonequilibrium protocols, although such processes typically reduce efficiency due to irreversibility. Driving schemes such as shortcuts to adiabaticity[16–18] fast-forward a quantum adiabatic evolution in finite time, but their exact implementation can be challenging[19] and aims for the same efficiency of adiabatic thermal cycles.

Designing quantum heat engines utilizing many-body systems as a working medium is necessary for their scaling[17,20] and paves the way to harness a wide variety of phenomena without a single-particle counterpart, including quantum statistics[21,22], interparticle interactions[9,23–25], and critical phenomena[26]. As many-body systems generally thermalize, the working medium follows equilibrium states if subject to slow operations. Hence, most previous studies have considered working medium at thermal equilibrium. Yet, several many-body systems feature constraints that forbid canonical thermalization[27] and present genuine prethermal phases in which quantum simulators[28,29] could perform reversible operations. A natural question is whether this scenario could be advantageous in increasing the performance of quantum heat engines. An important precedent in this regard is the study of quantum heat engines that harness many-body localization[30].

This article investigates the impact of prethermalization[27] in extended many-body systems on the engine efficiency, revealing that its advantage, relative to thermalizing working media, is universally determined by the temperatures of the thermal baths. In particular, prethermal heat engines that operate at negative temperatures exhibit a universal efficiency boost.

We consider Otto cycles operating between two thermal baths and performing work through a tunable parameter $\chi$. Without altering the baths, we compare prethermalization against thermalization during the adiabatic strokes, highlighting the role of the many-body working medium, whereas previous works focused on prethermal baths in few-body engines[31]. We consider the following strokes depicted in Fig. 1:

[1]Department of Mathematics, Technical University of Munich, CIT, Garching, Germany. [2]Physics Department, Technical University of Munich, TUM School of Natural Sciences, Garching, Germany. [3]Department of Physics and Materials Science, University of Luxembourg, Luxembourg, Luxembourg. [4]Donostia International Physics Center, San Sebastián, Spain. [5]Munich Center for Quantum Science and Technology (MCQST), München, Germany. ✉ e-mail: alvise.bastianello@tum.de

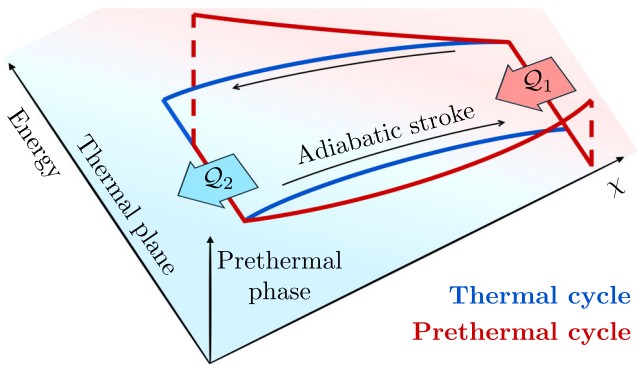

**Fig. 1 | Thermal vs prethermal Otto cycles.** The thermodynamic ensemble (thermal plane) describing the adiabatic strokes of a thermalizing working medium is described by a few conserved charges, including the energy, and the control parameter $\chi$. Prethermal working matter is characterized by a larger number of charges, exploring prethermal phases.

(i)   Adiabatic transformation. The system starts at thermal equilibrium with the bath, then it evolves in isolation, slowly changing $\chi$ and possibly exploring prethermal phases. It exchanges work $\mathcal{W}_1$, but not heat, and the entropy is conserved.

(ii)  Isochore transformation. At constant $\chi$, the system reaches thermal equilibrium with the second bath exchanging heat $\mathcal{Q}_1$, but not work.

(iii) The cycle is closed with another adiabatic stroke exploring a prethermal phase, exchanging work $\mathcal{W}_2$, and a isochoric transformation, exchanging heat $\mathcal{Q}_2$.

The system is governed by a Hamiltonian of the form $H(\chi) = H_0(\chi) + \epsilon V$, where the perturbation governed by the small parameter $\epsilon$ governs prethermalization. Specifically, we assume the unperturbed Hamiltonian $H_0(\chi)$ features $N$ conserved charges $\{Q_j\}_{j=1}^N$, which may depend on $\chi$, and satisfy $[Q_j, H_0(\chi)] = 0$. The small perturbation $\epsilon V$ is assumed to break some charges and deforms the others $Q_j \to Q_j' = Q_j + \delta Q_j(\epsilon)$, in such a way that $[Q_j', H(\chi)] = 0$ for $j \in \{1, \ldots, N' < N\}$. This is a fairly common scenario. For example, resonant tunneling in multicomponent ultracold gases[32] or tunneling in adjacent superfluids[33] breaks inter-species particle number conservation, Floquet-engineered Hamiltonians showcase symmetries broken at finite driving frequencies[34], and integrable systems[35,36] have infinitely many conservation laws broken by perturbations.

Isolated quantum systems with weakly-broken conserved charges undergo prethermalization: after a fast timescale $t_{pth}$ decided by microscopic processes[37], the system relaxes to the maximally-entropic state compatible with the conservation laws of the unperturbed Hamiltonian $H_0$. This is the prethermal phase. Then, on a slow time scale $t_{th} \gg t_{pth}$ usually polynomially growing in $\epsilon^{-1}$ [38–40], relaxation to a steady state determined by the reduced set of charges is observed. This is the thermal phase, where only few charges like the Hamiltonian and the particle number are conserved. For commuting charges, the two phases are described by generalized Gibbs ensemble (GGE) $\hat{\rho} = Z^{-1}e^{-\sum_j \beta_j Q_j}$ [35,41] where the appropriate charges and generalized inverse temperatures $\beta_j$ are considered. Although the GGE excludes certain nonergodic mechanisms, such as many-body localization[30,42] and fractons[43], it remains very general.

To highlight the impact of conservation laws only, we consider adiabatic processes where the adiabatic strokes follow the appropriate GGE, being it prethermal ($t_{pth} \ll \chi/(\mathrm{d}\chi/\mathrm{d}t) \ll t_{th}$) or thermal ($t_{th} \ll \chi/(\mathrm{d}\chi/\mathrm{d}t)$), and compare the two cases in the small $\epsilon$ limit. This adiabaticity requirement is much less stringent than quantum adiabaticity[44], whose time scale diverges in the absence of an energy gap, as is common in

the thermodynamic limit. Our focus is on the cycle's efficiency, namely the ratio between the total work $\mathcal{W} = \mathcal{W}_1 + \mathcal{W}_2$ and the absorbed heat $\mathcal{Q}_{abs} = \max(\mathcal{Q}_1, \mathcal{Q}_2)$

$$\eta = \mathcal{W}/\mathcal{Q}_{abs}. \qquad (1)$$

We unveil a universal efficiency enhancement: Thermalizing medium is more efficient for a positive bath temperature, whereas prethermalization is convenient at a negative temperature. This holds whenever all the charges conserved by the thermalizing dynamics, with the exception of the Hamiltonian $H_0(\chi)$, are independent of the control parameter $\chi$, while the whole set of prethermal charges can depend on $\chi$. We provide analytical proof for infinitesimal cycles on the basis of general thermodynamic inequalities without any assumption on the number of conservation laws, the form of interactions, or the dimensionality of the system. We furthermore demonstrate our findings using finite cycles with integrable systems, i.e., minimal interacting one-dimensional models featuring infinitely many conservation laws, amenable to many-body analytical computations far from equilibrium[35,45–47]. An interaction-driven quantum Otto cycle has been experimentally realized in a three-dimensional atomic cloud across the BEC-BCS crossover[9], and nearly-integrable variants are also possible[23]. However, realizing negative temperatures requires Hamiltonians with a finite maximum energy: this is not possible in continuous systems, but it is conceivable in experiments on a lattice[14,48,49]. As a proof of concept, we concretely discuss how our findings can be probed in state-of-the-art quantum gas microscopes, realizing integrable spin chains[50] with tunable integrability-breaking perturbations. In this context, we discuss how to engineer negative temperature states and measure the work done during the (pre)thermal adiabatic strokes.

## Results
### The adiabatic flow equations

We begin studying the evolution of the state during the adiabatic strokes. In the limit of slow changes of the control parameter $\chi$, the system instantaneously follows a GGE with evolving generalized temperatures, which we conveniently arrange in a vector $\{\beta_j\}_j \to \boldsymbol{\beta}$. As advanced in the introduction, we consider the limit where the perturbation breaking conservation laws is infinitesimal $\epsilon \to 0$, such that its effect is solely to break certain charges, without substantially altering the remaining ones.

We derive the flow equations that govern the adiabatic strokes in the prethermal states, since the thermal case follows similarly. To this end, it is convenient to approximate the smooth evolution as a sequence of sudden increments $\chi \to \chi + \mathrm{d}\chi$ separated by a waiting time $\mathrm{d}t$. The adiabatic limit $\mathrm{d}\chi/\mathrm{d}t \to 0$ is then taken, considering a large waiting time, in such a way that the system prethermalizes to the new generalized inverse temperatures $\boldsymbol{\beta}^{pth}(\chi + \mathrm{d}\chi)$. Let $Q_j(\chi)$ be the parametrically $\chi$ − dependent charge, and $\langle \ldots \rangle_{\chi,\boldsymbol{\beta}}$ be the expectation value in the prethermal state at $\chi$. The parameters $\boldsymbol{\beta}^{pth}$ are determined by charge conservation $\langle Q_j(\chi + \mathrm{d}\chi) \rangle_{\chi, \boldsymbol{\beta}^{pth}(\chi)} = \langle Q_j(\chi + \mathrm{d}\chi) \rangle_{\chi + \mathrm{d}\chi, \boldsymbol{\beta}^{pth}(\chi + \mathrm{d}\chi)}$. To linear order, one gets the flow equations

$$C_{pth}\partial_\chi \boldsymbol{\beta}^{pth} + A_{pth}\boldsymbol{\beta}^{pth} = 0, \qquad (2)$$

where the $\chi$ − dependence is omitted to ease the notation. Above, we defined the static covariance matrix as the connected charge-charge correlators $[C_{pth}]_{i,j} = \langle Q_i Q_j \rangle_c$, and the susceptibility matrix $[A_{pth}]_{i,j} = \langle Q_i \partial_\chi Q_j \rangle_c$. Here we defined the connected correlators as $\langle \mathcal{O}_1 \mathcal{O}_2 \rangle_c \equiv \langle \mathcal{O}_1 \mathcal{O}_2 \rangle - \langle \mathcal{O}_1 \rangle \langle \mathcal{O}_2 \rangle$. In addition, Eq. (2) implies the adiabatic evolution of the charges

$$\partial_\chi \langle Q_j \rangle = \langle \partial_\chi Q_j \rangle. \qquad (3)$$

The derivation of Eqs. (2) and (3) is reported in Methods. The thermal flow equations are identical to Eq. (2), restricted to the proper conserved charges and inverse temperatures $\boldsymbol{\beta}^{\text{th}}$.

To relate the flow equations (2) and the conventional notion of adiabaticity based on entropy, it is instructive to define the free energy associated with the GGE through the partition function as $\mathcal{F} = -\log \text{Tr}\left[e^{-\sum_j \beta_j Q_j}\right]$, from which the entropy $S$ is defined through canonical thermodynamic identities $\mathcal{F} = \sum_j \beta_j \langle Q_j \rangle - S$. By taking the time derivative of the last expression and comparing it with $\frac{d\mathcal{F}}{dt}$ obtained from the partition function, one reaches

$$\frac{dS}{dt} = \frac{d\chi}{dt} \sum_j \beta_j \left( \partial_\chi \langle Q_j \rangle - \langle \partial_\chi Q_j \rangle \right). \tag{4}$$

Notice that Eq. (3) implies $\frac{dS}{dt} = 0$, and it follows from the flow equations (2) that the adiabatic strokes are reversible in the conventional sense, regardless of whether the working medium is described by a thermal or prethermal phase. Therefore, both adiabatic strokes done with a thermalizing and prethermalizing medium belong to the class of reversible operations: our goal is now to understand which choice benefits the engine's efficiency. Eq. (2) is difficult to solve as it is highly nonlinear, since the expectation values evolve with the complex many-body state. Further progress can be made in generic infinitesimal cycles and in integrable models where the static covariance matrix $A$ and the susceptibility matrix $C$ can be analytically computed.

## Universal efficiency boost in infinitesimal cycles

Although infinitesimal cycles cannot be reliably used to deduce the behavior of finite cycles, they already provide a good indication. Furthermore, we found infinitesimal cycles to be amenable of analytical universal results. Our findings are universal in the sense that they rely solely on general thermodynamic inequalities, without making any assumptions about the number of conserved quantities, the form of interactions, or the dimensionality of the system. Hence, they have the broadest applicability.

We consider two thermal baths at $\boldsymbol{\beta}$ and $\boldsymbol{\beta} + \delta\boldsymbol{\beta}$, and the two strokes running from $\chi$ to $\chi + \delta\chi$. The change in internal energy during a stroke is obtained by expanding the integrated equation (3), where we consider the Hamiltonian as the conserved charge $\Delta\langle H \rangle = \int_\chi^{\chi+\delta\chi} d\chi' \langle \partial_{\chi'} H \rangle_{(\chi', \boldsymbol{\beta}(\chi'))}$. We find

$$\Delta\langle H \rangle = \delta\chi \langle \partial_\chi H \rangle + \frac{(\delta\chi)^2}{2} \left[ \partial_\chi \langle \partial_\chi H \rangle + \partial_\chi \beta_j \partial_{\beta_j} \langle \partial_\chi H \rangle \right], \tag{5}$$

where repeated indices are summed and terms $\mathcal{O}(\delta\chi)^3$ are neglected. Expectation values are computed on the initial thermal state, and the choice of flow equations (2) determines the evolution through thermal or prethermal states. Work is obtained by summing the two contributions from $\boldsymbol{\beta}$ and $\boldsymbol{\beta} + \delta\boldsymbol{\beta}$, and expanding in $\delta\boldsymbol{\beta}$. Notice that the flow equation (2) does not affect the first order $\sim \delta\chi$ contribution, but only the second order $\sim \delta\chi^2$. Therefore, differences in the two cycles appear at order $\sim \delta\chi^2$. Combining Eq. (2) and the identity $\partial_{\beta_j} \langle \partial_\chi H \rangle = -\langle Q_j \partial_\chi H \rangle_c$, the work difference $\delta\mathcal{W} \equiv \mathcal{W}^{\text{pth}} - \mathcal{W}^{\text{th}}$ is determined by the covariance and susceptibility matrices. The result is further simplified if all charges in the thermal state, except for the Hamiltonian itself, are $\chi$ − independent. In fact, $A_{i,j}$ vanishes for all indices $j$ of the charges of the thermalizing dynamics, with the exception of the Hamiltonian itself. $A_{i,j}$ may be non-zero for other prethermal charges, but these do not couple to the initial bath $\boldsymbol{\beta}$ and lead to

$$\delta\mathcal{W} = -\beta(\delta\chi)^2 \left[ A_{\text{pth}}^T C_{\text{pth}}^{-1} A_{\text{pth}} - A_{\text{th}}^T C_{\text{th}}^{-1} A_{\text{th}} \right]_{H,H}, \tag{6}$$

where $[\ldots]_{H,H}$ denotes the diagonal element in the Hamiltonian direction, and $\beta$ is the canonical inverse temperature. The sign of Eq. (6) crucially depends on $\beta$, with $\delta\mathcal{W}$ having a sign opposite to $\beta$. In fact, the matrix product in Eq. (6) is reformulated within hydrodynamic projections[51,52] (see Methods) as the norm of a vector subtracting its projection onto a smaller subspace, which is always positive. After having considered the work difference, we now focus on the absorbed heat and note that both the thermal and prethermal strokes start with the same energy $\langle H \rangle_{\chi,\boldsymbol{\beta}}$. After the first adiabatic stroke, they will have reached internal energies $\langle H \rangle_{\chi,\boldsymbol{\beta}} + \mathcal{W}_1^{\text{th, pth}}$. The absorbed heat is then computed as the difference in internal energy determined by the second bath, identical for the two working media, and the end of the adiabatic protocol $\mathcal{Q}_{\text{abs}}^{\text{th, pth}} = \langle H \rangle_{\chi+\delta\chi,\boldsymbol{\beta}+\delta\boldsymbol{\beta}} - \langle H \rangle_{\chi,\boldsymbol{\beta}} - \mathcal{W}_1^{\text{th, pth}}$. Inspection of Eq. (5) shows that $\mathcal{W}_1^{\text{pth}} - \mathcal{W}_1^{\text{th}} = \mathcal{W}_2^{\text{pth}} - \mathcal{W}_2^{\text{th}} = \frac{1}{2}\delta\mathcal{W}$, whence it follows that $\mathcal{Q}_{\text{abs}}^{\text{th}} - \mathcal{Q}_{\text{abs}}^{\text{pth}} = \frac{1}{2}\delta\mathcal{W}$. Therefore, the relative efficiency can be written as $\frac{\eta^{\text{pth}}}{\eta^{\text{th}}} = \left(1 + \frac{\delta\mathcal{W}}{\mathcal{W}^{\text{th}}}\right)\left(1 - \frac{\delta\mathcal{W}}{2\mathcal{Q}_{\text{abs}}^{\text{th}}}\right)^{-1}$ and it is thus greater than one for $\delta\mathcal{W} > 0$, which implies $\eta^{\text{pth}} > \eta^{\text{th}}$. We conclude that infinitesimal Otto cycles operating with thermal medium are more efficient than prethermal ones at positive temperatures, whereas the opposite holds for negative $\beta$.

## Prethermal finite cycles in integrable models

We next explore the persistence of the efficiency inequality for finite cycles in nearly integrable models[38,53]. These systems are realized in cold atoms[36], have infinitely many conserved charges, are strongly interacting, and are yet amenable to exact analytical computations, making them ideal candidates for our scope. In GGEs, the expectation values of charges, the covariance and susceptibility matrices can be computed exactly within the framework of Thermodynamic Bethe Ansatz (TBA)[54]; see Methods. We focus on two prototypical examples: the Ising model

$$H_{\text{Ising}} = -\sum_j \left( \sigma_{j+1}^x \sigma_j^x + h\sigma_j^z \right), \tag{7}$$

and the XXZ spin chain

$$H_{\text{XXZ}} = -J\sum_j \left( \sigma_{j+1}^x \sigma_j^x + \sigma_{j+1}^y \sigma_j^y + \Delta\sigma_{j+1}^z \sigma_j^z \right), \tag{8}$$

where $\sigma_j^{x,y,z}$ are canonical Pauli matrices acting on the $j$-th site. Notice that for the sake of simplicity, we work in dimensionless energy units. We consider homogeneous infinite systems and use as control parameters the magnetic field $h \to \chi$ and the anisotropy $\Delta \to \chi$ respectively. The Ising model is a canonical example of an integrable system, equivalent to noninteracting fermions with momentum $\lambda$ and dispersion $e(\lambda) = 2\sqrt{(\cos\lambda - h)^2 + \sin^2\lambda}$; see Supplementary Discussion 1 for details. We choose it primarily for pedagogical reasons[55] and for its broad relevance from non-equilibrium physics[37,56] to quantum engines[57–59]. The XXZ chain is a paradigmatic example of an interacting integrable model[60] implemented in quantum simulators[50,61–65]; see Supplementary Discussion 2 for details of its thermodynamics. In the Ising chain, many-body eigenstates can be described as a gas of free fermionic quasiparticles with momentum density $\rho(\lambda)$ and extensive energy $\langle H_{\text{Ising}} \rangle = L\int d\lambda\, e(\lambda)\rho(\lambda)$, with $L$ the system size (See Supplementary Discussion)[55], where we removed the ground state energy. Choosing the function $\rho(\lambda)$ is equivalent to fix the parameters $\beta_j$ of the GGE[35], see Methods and Supplementary Discussion 1. On thermal states, it has Fermi-Dirac statistics $\rho(\lambda) = \frac{1}{2\pi}(e^{\beta e(\lambda)} + 1)^{-1}$. The same picture holds in the XXZ chain, albeit interactions dress the excitations and deform thermal distributions through nonlinear integral

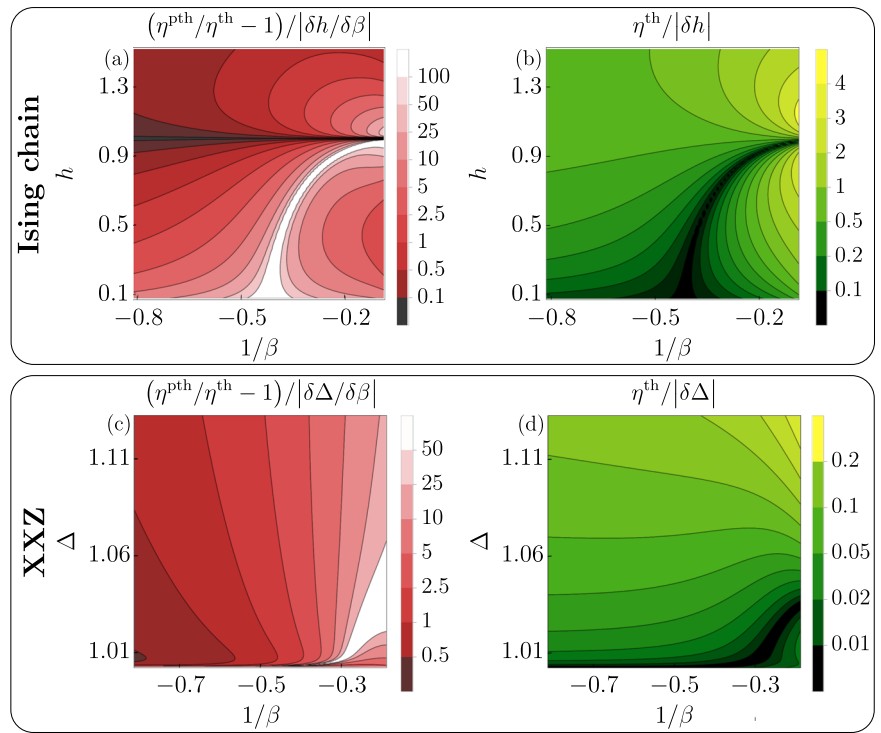

**Fig. 2 | Thermal vs Prethermal infinitesimal Otto cycles in integrable models.** We show the relative efficiency of thermal-prethermal infinitesimal Otto cycles and the thermal infinitesimal efficiency in the Ising (7) (**a**, **b**) and antiferromagnetic $J = -1$ XXZ (8) (**c**, **d**) chains with average magnetization $\langle \sigma^z \rangle = 0.45$. We use as tunable parameters the magnetization $h$ and anisotropy $\Delta$, respectively. To plot efficiencies independent of the cycle size, we focus on skewed cycles where the absorbed heat is much larger than the work, $\mathcal{W} \ll \mathcal{Q}_{abs}$, or equivalently $|\delta\chi| \ll |\delta\beta|$. In this regime, the efficiency scales as $\eta \sim \delta\chi$ and $\eta^{pth}/\eta^{th} - 1 \sim \delta\chi/\delta\beta$, and the data is rescaled to factor out the explicit dependence on the size of the infinitesimal stroke. Explicit formulas are obtained from Eq. (5) and reported in Supplementary Discussion. Notice that the regions of large relative efficiency are related to regions of vanishing thermal efficiency.

equations (See Supplementary Discussion). The flow equations Eq. (2) can be reformulated in the quasiparticle basis within Generalized Hydrodynamics (GHD)[45–47,66], a non-perturbative framework for nearly integrable systems. In homogeneous systems with slow-varying interactions, the GHD equations are[67]

$$\partial_\chi \rho(\lambda) + \partial_\lambda (F^{eff}(\lambda)\rho(\lambda)) = 0. \tag{9}$$

The effective force $F^{eff}$ captures the many-body effect of varying interactions, and it vanishes for noninteracting models such as the Ising chain; see Methods for details. In Fig. 2, we show the efficiencies for infinitesimal cycles for Ising and XXZ chains, in a broad spectrum of parameters. Thermal states in the Ising chain are determined solely by the energy. By contrast, in the XXZ model we consider integrability-breaking perturbations that preserve the total magnetization $\sum_j \sigma_j^z$, resulting in thermal states with two conserved quantities. Notice that the total magnetization does not depend on the anisotropy, i.e., our control parameter. As expected from our general argument, pre-thermal states are more efficient than thermal ones at negative temperatures. In Fig. 3, we consider exemplificative cases of finite cycles numerically solving Eq. (9) (See Supplementary Discussion). For finite strokes, the gain becomes comparable with the efficiency itself, and is generally larger for the Ising chain. This is due to the fact that thermal states in the XXZ chain have two conserved charges rather than one as in the Ising case, allowing them to be closer to the GGE. More details are provided in Supplementary Discussion 2.

## Quantum engine simulators
The perfect isolation of quantum simulators that makes long-time coherent dynamics and negative temperature possible hampers quantum engines requiring heat exchanges with thermal baths[23,28,29,68].

Nonetheless, current platforms can already probe the two adiabatic strokes of the Otto cycle separately. The XXZ chain (8) with positive spin exchange $J > 0$ is realized in one-dimensional gases at unit filling[69], encoding the $z$ − direction of the spin in two hyperfine levels. Convenient platforms for our scopes are Lithium-based implementations in optical lattices[61], where $\Delta$ is tunable thanks to a Feschbach resonance, and quantum gas microscopes[50] due to their ability to conduct single-site measurements and operations. The current XXZ chain quantum microscope with Rubidium atoms[50] lacks a Feshbach resonance, fixing $\Delta \simeq 1$. However, quantum microscopes with Lithium are also available[70], and could combine the advantages of the two platforms in the near future. The tunable transverse confinement efficiently breaks integrability, interpolating between a one-dimensional and a ladder geometry[50]. The use of thermal baths at negative temperature generally requires engineering for their preparation. In this regard, negative temperatures can be realized by selectively exciting spins in high-energy configurations and evolving them in the presence of integrability-breaking perturbations that induce thermalization. For example, such states for the XXZ chain in the ferromagnetic phase could be an antiferromagnetic spin arrangement. Atom imaging provides snapshots of the $z$ − magnetization, from which arbitrary $zz$ correlations can be obtained[50]. Directly probing the energy requires measurements in the other spin directions as well, but adiabatic operations conveniently give direct access to energy differences through integration of Eq. (3) which, for the case of a tunable $\Delta$, requires measuring $\langle \sigma_{j+1}^z \sigma_j^z \rangle$ only. In the absence of a tunable $\Delta$, time-dependent magnetic traps $H_{XXZ} + \sum_j B_j(t)\sigma_j^z$ can be used to exert work, as suggested in refs. 20,29,69. Indeed, smooth traps break integrability weakly, resulting in long-lived prethermal states[53,71,72]. This possibility, however, pivots (pre)thermalization timescales to the trap's size: on the typical sizes of a few tens of spins, a conservative estimation

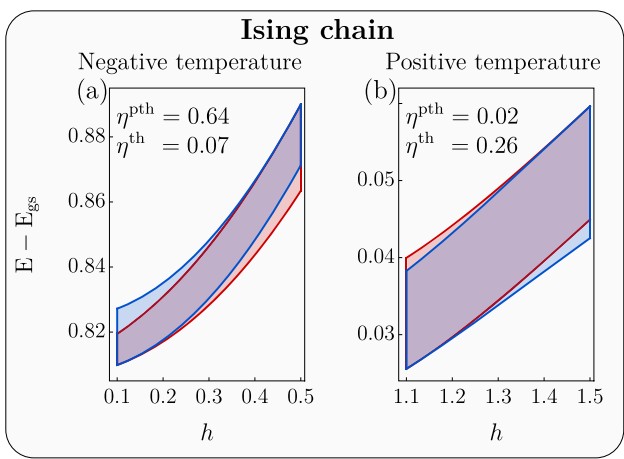
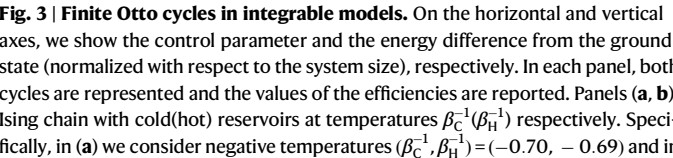
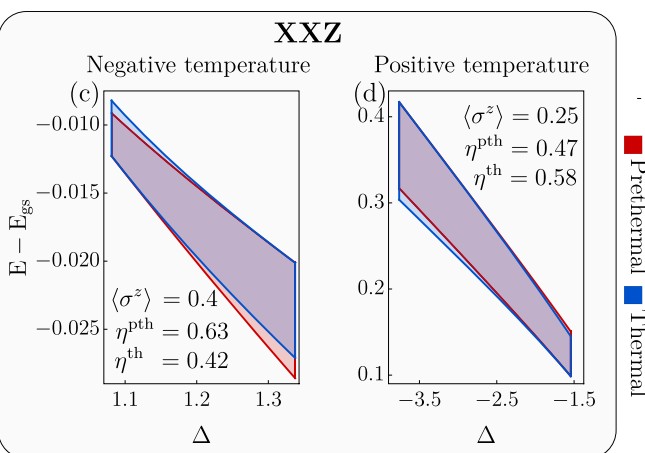

**Fig. 3 | Finite Otto cycles in integrable models.** On the horizontal and vertical axes, we show the control parameter and the energy difference from the ground state (normalized with respect to the system size), respectively. In each panel, both cycles are represented and the values of the efficiencies are reported. Panels (**a**, **b**): Ising chain with cold(hot) reservoirs at temperatures $\beta_C^{-1}(\beta_H^{-1})$ respectively. Specifically, in (**a**) we consider negative temperatures $(\beta_C^{-1}, \beta_H^{-1}) = (-0.70, -0.69)$ and in

(**b**) positive temperature $(\beta_C^{-1}, \beta_H^{-1}) = (0.30, 0.48)$. Panels (**c**, **d**): analog cases for the XXZ chain (8) with $J = -1$ and fixed magnetization $\langle \sigma \rangle$. The temperatures of the baths in (**c**) and (**d**) are $(\beta_C^{-1}, \beta_H^{-1}) = (-0.175, -0.150)$ and $(\beta_C^{-1}, \beta_H^{-1}) = (0.5, 2.0)$. These examples show that the general conclusions for infinitesimal cycles remain valid also for finite operations. A quantitative measure of the GGE's departure from canonical equilibrium and further cases are reported in Supplementary Discussion.

suggests timescales of various tens of spin-exchange times, challenging the present coherence time. Instead, (pre)thermalization after homogeneous quenches in $\Delta$ requires ~5 spin-exchanges[73], and thus is more convenient.

## Discussion

In this article, we unveiled the universal impact of conservation laws and prethermalization on quantum engines. By focusing on Otto cycles, we have established general thermodynamic inequalities showing how the relative efficiency of small cycles with thermal or prethermal working medium is entirely determined by the baths' temperature. Specifically, a thermal working medium is more efficient at positive temperatures, whereas prethermal media enhance the engine efficiency at negative temperatures. Although the use of negative temperatures generally requires population inversion, one can anticipate scenarios in which the benefits of their use outperform their cost, if not fundamentally, in practice. We focused on integrable models as a concrete case of study, showing the persistence of our conclusions beyond small cycles, where Generalized Hydrodynamics provides exact quantitative insight into far-from-equilibrium quantum matter. Our findings are of direct relevance to state-of-the-art quantum simulators. While we thoroughly discussed the realization of the XXZ model in quantum gas microscopes, other platforms such as Rydberg atoms in optical tweezers[74], superconducting qubits[75], and trapped ions[76] could also be employed. Fermi-Hubbard quantum microscopes with Feschbach resonances, offer another natural platform[70] in the context of nearly-integrable models. However, as shown by our analysis of infinitesimal cycles, our results are of broad relevance beyond integrability itself and apply generally whenever a conservation quantity can be selectively broken and are thus of broad experimental relevance. It is worth emphasizing that, albeit we focused on quasi-static protocols, (pre)thermalization time scales are driven by microscopic processed[37–40] that are usually fast compared with other typical scales in experiments[36], suggesting our results can also be predictive for finite-time protocols. Interesting future directions involve exploring the consequences of nonthermal baths, which may be realized by coupling different portions of isolated quantum many-body systems[29], and considering driving protocols in finite time, and their role on the tradeoff between the efficiency and power, and (pre)thermalization.

## Methods

### Derivation of the flow equations

We derive the flow equations (2), which govern the adiabatic evolution. From the partition function $Z(\chi, \boldsymbol{\beta}) = \mathrm{Tr}\left[e^{-\sum_i \beta_i Q_i(\chi)}\right]$, one has the standard thermodynamic equalities $\langle Q_j(\chi) \rangle_{\chi, \beta_i} = -\partial_{\beta_i} \log Z(\chi, \boldsymbol{\beta})$, $\langle \partial_\chi Q_j(\chi) \rangle_{\chi, \beta_i} = -\partial_\chi \log Z(\chi, \boldsymbol{\beta})$, while the second mixed derivatives give connected correlation functions. Promoting $\boldsymbol{\beta}$ to be $\chi$ − dependent in the adiabatic stroke, and imposing charge conservation $\langle Q_j(\chi + \delta\chi) \rangle_{\chi, \boldsymbol{\beta}(\chi)} = \langle Q_j(\chi + \delta\chi) \rangle_{\chi + \delta\chi, \boldsymbol{\beta}(\chi + \delta\chi)}$ to first order in $\delta\chi$, the flow equations Eq. (2) immediately follow. Similarly, the generic variation of a charge can also be obtained with changes in $\chi$ and $\beta_i$, leading to $\delta\langle Q_j \rangle = \delta\chi \langle \partial_\chi Q_j \rangle - \sum_i C_{ji} \delta\beta_i - \delta\chi \sum_i A_{ji} \beta_i$. Imposing that $\boldsymbol{\beta}$ evolves with the flow equations, $\partial_\chi \langle Q_j \rangle = \langle \partial_\chi Q_j \rangle$ follows.

### Hydrodynamic projections

Historically, hydrodynamic projections[51,52] have been introduced to isolate the slow, long-wavelength dynamics of a many-body system by projecting onto conserved quantities. We use this framework to conveniently rewrite Eq. (6) and make its sign explicit. Due to the infinitesimal nature of the perturbation, the leading-order effect is the suppression of certain conserved quantities, rather than their deformation. Consequently, the set of conserved quantities under thermalizing dynamics spans a strict subspace of those preserved in the prethermal regime. One introduces a scalar product in the vector space of the observable through their connected correlator $\langle \mathcal{O}_1 | \mathcal{O}_2 \rangle \equiv \langle \mathcal{O}_1 \mathcal{O}_2 \rangle_c$. We define the projector on the conserved charges of the (pre)thermal dynamics as $\mathbb{P}^{\mathrm{pth(th)}} = \sum_{i,j}^{N(N')} [C_{\mathrm{pth(th)}}^{-1}]_{i,j} |Q_i\rangle \langle Q_j|$, where the inverse static covariance matrix $C^{-1}$ is introduced for a properly normalized projection $\mathbb{P}^{\mathrm{pth(th)}}|Q_j\rangle = |Q_j\rangle$. In this language, Eq. (6) is rewritten as the difference of the norm of a vector projected on different subspaces

$$\delta\mathcal{W} = -\beta(\delta\chi)^2 \left[ \langle \partial_\chi H | \mathbb{P}^{\mathrm{pth}} | \partial_\chi H \rangle - \langle \partial_\chi H | \mathbb{P}^{\mathrm{th}} | \partial_\chi H \rangle \right]. \quad (10)$$

Since the space of thermal conserved charges is included in the prethermal one $[\langle \partial_\chi H | \mathbb{P}^{\mathrm{pth}} | \partial_\chi H \rangle - \langle \partial_\chi H | \mathbb{P}^{\mathrm{th}} | \partial_\chi H \rangle \geq 0$, proving the sign of $\delta\mathcal{W}$ depends only on $\beta$.

## Integrable systems

Multiparticle scattering events in integrable models are factorized into two-by-two elastic scattering events, entirely parametrized by their scattering phase. In interacting integrable models, the rapidity $\lambda$ and root density $\rho(\lambda)$ generalize the momentum and momentum density of the free systems, respectively. An additional degree of freedom labeling quasiparticles of different species is present in several cases, such as in the XXZ model[54], and is here omitted for brevity. The expectation value of the conserved charges takes the form $\frac{1}{L}\langle Q_i \rangle = \int d\lambda\, q_i(\lambda)\rho(\lambda)$, with $q_i(\lambda)$ being the charge eigenvalue. The scattering phase $\Theta(\lambda - \lambda')$ between two excitations of rapidity $\lambda$ and $\lambda'$ depends on the rapidity difference. Thermal states and GGEs are described within the thermodynamic Bethe ansatz (TBA) framework[54]. More precisely, the state is parametrized by nonlinear integral equations

$$\varepsilon(\lambda) = \sum_i \beta_i q_i(\lambda) + \int \frac{d\lambda'}{2\pi} \varphi(\lambda - \lambda') \log(1 + e^{-\varepsilon(\lambda')}), \tag{11}$$

where the scattering kernel is defined as $\varphi(\lambda) \equiv \partial_\lambda \Theta(\lambda)$. The pseudoenergy $\varepsilon(\lambda)$ parameterizes the state through the filling function $\vartheta(\lambda) = (1 + e^{\varepsilon(\lambda)})^{-1}$, which is then connected to the root density as $\rho(\lambda) = \vartheta(\lambda)\frac{(\partial_\lambda p)^{dr}}{2\pi}$, with $p(\lambda)$ the momentum of a quasiparticle. In general, the dressing of a bare quantity $\tau(\lambda)$ is given as a solution of the linear integral equation $\tau^{dr}(\lambda) = \tau(\lambda) - [\varphi * \vartheta \tau^{dr}](\lambda)$. For brevity, we define the convolution $[\varphi * \tau](\lambda) = \int d\lambda' \varphi(\lambda - \lambda')\tau(\lambda')$. The static covariance matrix is analytically determined as $\langle Q_i Q_j \rangle_c = -\partial_{\beta_i}\langle Q_j \rangle$ (See Supplementary Discussion)

$$\frac{1}{L}\langle Q_i Q_j \rangle_c \simeq^{L \to \infty} \int d\lambda\, q_i^{dr} \rho(1 - \vartheta) q_j^{dr}, \tag{12}$$

where $\simeq$ denotes equality in the thermodynamic limit. The susceptibility matrix follows from $\langle Q_i \partial_\chi Q_j \rangle_c = -\partial_{\beta_i}\langle \partial_\chi Q_j \rangle$, where $\langle \partial_\chi Q_j \rangle$ is computed by means of the Hellmann-Feynman theorem[67]

$$\frac{1}{L}\langle \partial_\chi Q_j \rangle \simeq \int d\lambda \left( \partial_\chi q_j \rho + \frac{1}{2\pi} \partial_\lambda q_j f^{dr} \vartheta \right), \tag{13}$$

where $f(\lambda) = -\partial_\chi p(\lambda) + [\partial_\chi \Theta * \vartheta(\partial_\lambda p)^{dr}](\lambda)$. Deriving Eq. (13), the susceptibility matrix follows (See Supplementary Discussion)

$$\frac{1}{L}\langle Q_i \partial_\chi Q_j \rangle_c \simeq \int d\lambda\, q_i^{dr} \rho(1-\vartheta) \left( f^{dr} \frac{(\partial_\lambda q_j)^{dr}}{(\partial_\lambda p)^{dr}} - \Lambda_j^{dr} \right) \tag{14}$$

where $\Lambda_i(\lambda) = -\partial_\chi q_i(\lambda) + [\partial_\chi \Theta * \vartheta(\partial_\lambda q_i)^{dr}](\lambda)$. With the covariance and susceptibility matrices at hand, the flow equations (2) are fully determined. In the prethermal case, rather than working with infinitely many charges, it is more convenient to move to a quasiparticle basis. Here, the flow equations are equivalent to the GHD equations (9)[67] with the effective force being $F^{eff}(\lambda) = f^{dr}(\lambda)/(\partial_\lambda p)^{dr}$, which can also be generalized to inhomogeneous setups. Notice that in non-interacting systems like the Ising model $\varphi = 0$, therefore, the equations greatly simplify. In Supplementary Discussion 2, we provide details for the general formulas for the XXZ spin chain in the easy-axis regime $|\Delta| > 1$. For $|\Delta| < 1$, the GHD equations for changing $\Delta$ are an open problem[67] that we do not address.

## Numerical methods

Finite cycles in integrable systems are obtained by numerically solving the TBA and GHD equations; see Supplementary Methods 1 for details. The integral equations are discretized and solved with standard methods. The GHD equation is solved using the method of characteristics at second order in time evolution[67]. The evolution along the thermal strokes is performed with the flow equations (2). We checked the convergence of our results with respect to the discretization in the

rapidity space, the number of quasiparticle species in the XXZ chain, and the integration time step. Raw data and a Mathematica code are available on Zenodo[77].

## Data availability

All the raw data are available on Zenodo[77].

## Code availability

Working Mathematica notebooks are available on Zenodo[77].

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

## Acknowledgements

We are indebited to Manuele Landini, Rosario Fazio, Alvaro Martin Alhambra and Herbert Spohn for useful discussion and comments on the manuscript. ABro acknowledges support from Deutsche Forschungsgemeinschaft (DFG, German Research Foundation) - TRR 352 - Project-ID 470903074. AdC acknowledges financial support from the Luxembourg National Research Fund (FNR Grant Nos. C24/MS/18940482/STAOpen). ABas acknowledges the support of the Deutsche Forschungsgemeinschaft (DFG, German Research Foundation) under the Germany's Excellence Strategy-EXC-2101-3990814868.

## Author contributions

A.Bro carried out the analytical computations and simulations, A.Bas devised the project's goal. A.Bas and A.C supervised the work. All authors contributed critically to the writing of the manuscript and the interpretation of results.

## Funding

## Competing interests

The authors declare no competing interests.
