## [Transparent Peer Review file · Nature Communications]

Universal efficiency boost in prethermal quantum heat engines at negative temperature

Corresponding Author: Dr Alvisè Bastianello

Version 0:

Reviewer comments:

Reviewer #1

(Remarks to the Author)

The manuscript tackles the problem of finding possible efficiency boosts in quantum heat engines using a quantum many-body system as a working fluid. In the last decade, there has been growing attention to demonstrating such kinds of advantages due to collective and/or genuine quantum effects in key thermodynamic tasks, like work extraction and energy conversion.

The goal of the manuscript is to compare the efficiency of the Otto cycle in two different settings, i.e., the prethermal and the thermal cycle respectively. During the adiabatic strokes of a prethermal cycle, the quantum state of the working medium evolves through a set of prethermal states that can be described in terms of an extensive number of conserved charges. On the other hand, during a conventional thermal cycle, a perturbation is acting on the system Hamiltonian to deform a finite number N^{\prime} of charges while breaking the others, thus leading to thermalization on a timescale set by the charge-breaking interaction.

The main result of the manuscript is that enhancements in the engine's efficiency due to prethermalization can be analytically proven. However, these performance boosts take place provided that the cold and hot reservoirs are kept at negative temperatures. On the contrary, for conventional, positive temperature reservoirs, thermal states are more convenient than prethermal ones. The result is proven analytically by means of exact generalized hydrodynamics theories, which are valid in the thermodynamic limit, and it is eventually verified by means of two paradigmatic Hamiltonians models of integrable and nearly integrable quantum many-body systems, i.e., the transverse-field Ising model and the XXZ model.

The results are remarkable, as they unveil a universal property of quantum many-body heat engines by means of the elegant hydrodynamic formalism. Also, due to the approach used, their validity is not limited to integrable and nearly integrable models, as they may find applications in all physical settings where conserved quantities can be defined. More importantly, the Authors envision suitable platforms based on quantum simulators, to experimentally probe their results. A further point of merit, the methodology employed is sound, and the overall quality of the supporting material is high, meaning that the relevant equations are accessible and relatively easy to work out.

On the other hand, while the manuscript provides useful insights into the thermodynamics of prethermalization, the significance of this work to the field of quantum heat engines and the novelty of its results may be controversial and difficult to assess.

Indeed, quite similar efficiency boosts have been experimentally verified in quantum heat engines working with negative temperature baths and with just a single degree of freedom used as the working medium, i.e., without the need for quantum many-body effects, see for instance Phys. Rev. Lett. 122, 240602 (2019). In addition, the correct evaluation of the engine's figures of merit when various kinds of non-thermal quantum resources are involved, along with their thermodynamic interpretation, have been proven as highly nontrivial tasks (see for instance Nat. Commun. 9, 165 (2018), or Phys. Rev. Research 2, 043302 (2020) for thermal machines using generalized resources).

It follows that, without further investigations, the impact of prethermalization in boosting the performance of quantum heat

engines may appear limited. Indeed, from the discussion of the results, it is not clear what kind of resource is responsible for the described performance boost, i.e., is it due to the dynamically generated prethermal states or rather to the nonthermal states of the baths? Another potential issue is that, in the field of nonequilibrium engines, the mere evaluation of efficiency may be not enough to assess the overall performance, see for instance Phys. Rev. Lett. 120, 190602 (2018). For instance, what is the effect of fluctuations in the power output? What about the entropy production?

On a minor note, below I list some points the Authors may want to address to further improve the quality of their presentation:

- 1- In the results section, when dealing with the difference in the heat exchanged during the isochore strokes, the Authors may want to expand the discussion to further clarify and/or report the full expression for the heat, from which the leading order expression could be worked out.
- 2- In Fig. 2, the Authors could comment on the low values of the relative efficiency in the case of the Ising chain, occurring around $h=1$. Is that related to the criticality of the quantum Ising model?
- 3- In Fig. 3, the insets are not fully clear. Is the black (orange) curve related to prethermal (thermal) cycle?
- 4- In the Methods section, when reporting on the hydrodynamic projections' techniques, some readers may miss the reason why the projector onto the N^{\prime} conserved charges is written in terms of the $Q_{\{j\}}$ s rather than the deformed charges.

(Remarks on code availability)

Reviewer #2

(Remarks to the Author)
See attachment

[Editorial Note: See end of file]

(Remarks on code availability)

Reviewer #3

(Remarks to the Author)

The authors show that prethermal quantum heat engines can be more efficient than thermal ones, for negative temperatures, while the opposite is true for positive temperatures. The authors start with a general formalism, and then consider specific examples of two integrable models.

The work is timely, rigorous, and the results seem important, and are applicable to a broad category of systems. My main comment is, the authors claim that advantage in efficiency exists for negative temperatures. However, negative temperatures are associated with ergotropy, and there already has been many works which show that negative temperatures enhance the performance of quantum engines. For example, see Phys. Rev. Lett. 112, 030602 (2014) and PRX Quantum 3, 040334 (2022). So given that it is already well known that negative temperatures boost the performance, I suggest the authors add a discussion on why they claim their results show something very novel. Does ergotropy play any role here?

Also given that negative temperature is the key here, I would suggest adding that in the title as well. But I leave this to the consideration of the authors.

My other minor comments are:

1. In the description of strokes, in page 1, above Fig. 1, the authors write that during the adiabatic stroke the system follows a thermal or prethermal state. However, when a Hamiltonian parameter is changed at a finite rate, in general the system goes out thermal equilibrium. So do the authors consider slowly changing Hamiltonian, or some control techniques, such as shortcuts to adiabaticity?
2. Similar comment for the isochoric stroke as well - the authors write that the system remains thermal. Do they mean the system remains at thermal equilibrium with the bath at the end of the stroke, or does the system remain at thermal equilibrium at all times, even during the stroke? I would expect the former, since I understand that the system is coupled to a bath at a constant temperature, and the heat flow brings the system to thermal equilibrium at the end of the stroke. The authors should clarify this point.
3. The authors consider N conserved quantities. Is N of the order of system size, or is it of the order of the Hilbert space dimension of the system?

(Remarks on code availability)

Version 1:

Reviewer comments:

Reviewer #1

(Remarks to the Author)

I was asked to reconsider the manuscript NCOMMS-25-28247A-Z, entitled "Universal efficiency boost in prethermal quantum heat engines at negative temperature".

Compared to its previous version, the title of the manuscript has been changed and a more accurate explanation of prethermal against thermal cycles has been reported. The manuscript has also benefited from the inclusion of a paragraph related to entropy production. Moreover, the main results have been emphasized, and additional details have been provided in the supplemental material.

Although the Authors replied to all my points, I confirm that the novelty and the actual significance of the presented results to the field of quantum heat engines (QHE) is difficult to assess. Nevertheless, the manuscript could more likely be relevant to the field of quantum simulators.

There is no doubt that prethermalization is a many-body effect with no single or few-body counterpart. However, the issue of whether the "[...] breakdown of conventional thermodynamics [...]" (i.e, the absence of thermalization) "[...] can be advantageous for a many-body quantum engine [...]" has been already addressed in the past, focusing on the very special case of ergodicity breaking in the MBL phase, see for instance Phys. Rev. B 99, 024203 (2019). Indeed, as reported in recent review papers on the subject, QHEs operating with many-body systems as their working medium have been studied for more than a decade. Moreover, as stated by the same Authors, when compared to previous proposals in the field, this engine delivers an amount of power that is close to zero.

While it is true that the thermodynamics of Generalized Gibbs Ensembles (GGE) has mainly been studied in the context of non-thermal baths, the prethermal state of the working medium, as stated by the same Authors, is modeled as a peculiar instance of GGE, which is adiabatically driven in time and sustains a number of conserved charges that can be deformed or broken due to the perturbation. Consequently, the occurrence of unconventional "efficiency boosts" due to the combined effect of nonequilibrium states of the working medium and of the baths cannot be considered as totally unexpected.

As already detailed in my first report, this manuscript stands out from previous works on many body QHEs mainly due to its rigorous proof of the properties of prethermal cycles. Indeed, the ingenious use of the perturbation-deformed charges provides a remarkable insight, i.e., adiabatically driven nonequilibrium prethermal states are more suitable to harness quantum resources (i.e., population-inverted negative temperature baths) when compared to thermalizing states, while the opposite holds true for more conventional resources, i.e., positive temperature baths. Due to the wise use of the powerful hydrodynamics formalism, the Authors prove this result to be universal, i.e., it can be extended to a wide class of interacting many-body systems. Eventually, the Authors also suggest several experimental platforms, based on ultracold atoms trapped in optical lattices, to probe their theoretical results.

Reviewer #2

(Remarks to the Author)

I want to thank the referees for their responses and for the modifications they implemented in the manuscript. I think they have properly addressed all of my concerns and that the current version of the manuscript is ready for publication.

Reviewer #3

(Remarks to the Author)

The authors have responded to the comments satisfactorily, and also modified the manuscript accordingly to highlight the main results and improve the clarity. In particular, the authors emphasize that the origin of the improvement in the performance for negative temperatures stems from the pre-thermal nature of the many-body quantum system. These results seem universal, and in my opinion can be a significant contribution in the topic of many-body quantum engines. However, preparing a bath at a negative temperature incurs a cost. Will these advantages change when that cost is taken into account?

I recommend publication of the manuscript once the authors have added a discussion on the above aspect.

Version 2:

Reviewer comments:

Reviewer #3

(Remarks to the Author)

The authors have considered my comments and modified the manuscript accordingly. I recommend publication of the manuscript in the present form.

Response to Referees for resubmission of NCOMMS-25-28247

Below, we provide a detailed point-by-point answer to the questions and suggestions raised by the Referees, and discuss the improvements we made in the manuscript to resolve these issues. We report quotes from the Referees's reports in *italic*, whereas our answers are reported in boxes.

Response to Referee 1

The manuscript tackles the problem of finding possible efficiency boosts in quantum heat engines using a quantum many-body system as a working fluid. In the last decade, there has been growing attention to demonstrating such kinds of advantages due to collective and/or genuine quantum effects in key thermodynamic tasks, like work extraction and energy conversion.

The goal of the manuscript is to compare the efficiency of the Otto cycle in two different settings, i.e., the prethermal and the thermal cycle respectively. During the adiabatic strokes of a prethermal cycle, the quantum state of the working medium evolves through a set of prethermal states that can be described in terms of an extensive number of conserved charges. On the other hand, during a conventional thermal cycle, a perturbation is acting on the system Hamiltonian to deform a finite number N' of charges while breaking the others, thus leading to thermalization on a timescale set by the charge-breaking interaction.

The main result of the manuscript is that enhancements in the engine's efficiency due to prethermalization can be analytically proven. However, these performance boosts take place provided that the cold and hot reservoirs are kept at negative temperatures. On the contrary, for conventional, positive temperature reservoirs, thermal states are more convenient than prethermal ones. The result is proven analytically by means of exact generalized hydrodynamics theories, which are valid in the thermodynamic limit, and it is eventually verified by means of two paradigmatic Hamiltonians models of integrable and nearly integrable quantum many-body systems, i.e., the transverse-field Ising model and the XXZ model.

The results are remarkable, as they unveil a universal property of quantum many-body heat engines by means of the elegant hydrodynamic formalism. Also, due to the approach used, their validity is not limited to integrable and nearly integrable models, as they may find applications in all physical settings where conserved quantities can be defined. More importantly, the Authors envision suitable platforms based on quantum simulators, to experimentally probe their results. A further point of merit, the methodology employed is sound, and the overall quality of the supporting material is high, meaning that the relevant equations are accessible and relatively easy to work out.

We thank the Referee for the accurate summary of our work and the overall positive assessment. We are pleased that the Referee shares our excitement, as they find our result remarkable, of broad applicability, sound, and of high quality.

On the other hand, while the manuscript provides useful insights into the thermodynamics of prethermalization, the significance of this work to the field of quantum heat engines and the novelty of its results may be controversial and difficult to assess.

Indeed, quite similar efficiency boosts have been experimentally verified in quantum heat engines working with negative temperature baths and with just a single degree of freedom used as the working medium, i.e., without the need for quantum many-body effects, see for instance *Phys. Rev. Lett.* 122, 240602 (2019). In addition, the correct evaluation of the engine's figures of merit when various kinds of non-thermal quantum resources are involved, along with their thermodynamic interpretation, have been proven as highly nontrivial tasks (see for instance *Nat. Commun.* 9, 165 (2018), or *Phys. Rev. Research* 2, 043302 (2020) for thermal machines using generalized resources).

We thank the Referee for pointing out these references, which are now properly mentioned in the manuscript, and encouraging us to further emphasize the novelty of our result and approach to previous literature. Based on the Referees' report, we have substantially improved the introduction to better emphasize our setting and keypoints, which we summarize here:

- Experimental advances in quantum simulators motivate us to explore *extended many-body* quantum systems, telling us apart from a large deal of previous studies focusing on few-body engines.
- Recent experimental and theoretical breakthroughs have generated a great excitement in nonequilibrium phases of matter evading canonical equilibrium, grouped within the phenomenon of prethermalization. Prethermalization is an *intrinsic many-body effect of the working medium*, and it is natural to wonder if the breakdown of conventional thermodynamics can be advantageous for a many-body quantum engine. To the best of our knowledge, this question has never been addressed in the past.
- Studying highly excited phases (e.g. finite-temperature thermal equilibrium, or far-from-equilibrium phases) in strongly interacting many-body systems is a daunting task, both numerically and analytically. For this reason, studies on equilibrium quantum many-body engines are scarce (see Refs. in the manuscript). From this perspective, it is even more remarkable that we managed to obtain analytical, rigorous results on nonthermal states. This has also been possible thanks to recent advances in the hydrodynamics of integrable models (GHD): Our work also has the merit of connecting two very active communities, namely the one working on quantum engines and experts in integrable models.
- Lastly, as the Referee points out, we envision concrete experimental platforms in quantum simulators where these finding can be tested, and possible collaborations with experimental groups are currently being discussed.

We are fully convinced that the unique combination of these groundbreaking achievements of our work greatly speaks in favor of its novelty, broad interest and future impact.

In light of the above observations, we also take this opportunity to comment further on the references mentioned by the Referee. We would like to first emphasize that our results are inherently many-body in nature, relying on the concept of prethermalization,

and thus cannot be established at the single-particle level, in which the suggested references are framed.

Please note that Phys. Rev. Lett. 122, 240602 (2019) reports an experimental demonstration of a quantum heat engine (QHE) with a single-qubit working medium. Reference Nat. Commun. 9, 165 (2018) discusses a photonic QHE utilizing an oscillator. Reference Phys. Rev. Research 2, 043302 (2020) has more direct relevance for our work, as it focuses on nonthermal resources for the baths coupled to two quantum dots. In contrast, we consider *many-body* working medium, whose prethermal phases are entirely responsible for the results we report. We properly acknowledge this work in line 56 of the revised manuscript.

It follows that, without further investigations, the impact of prethermalization in boosting the performance of quantum heat engines may appear limited. Indeed, from the discussion of the results, it is not clear what kind of resource is responsible for the described performance boost, i.e., is it due to the dynamically generated prethermal states or rather to the nonthermal states of the baths?

We thank the Referee for raising these important questions, which further motivate us to enhance the clarity of our manuscript. To better compare a quantum many-body engine working with thermal or prethermal medium, in our Otto cycle, we consider two conventional thermal baths (with the caveat of being able to realize negative temperature), which are kept the same when comparing the thermal and prethermal working medium. To this end, the only resource responsible for the performance boost is (pre)thermalization of the working medium generated during the adiabatic strokes, upon changing the control parameter χ . The introduction has been substantially improved to better convey this important fact, in particular at lines 50-57 of the resubmitted manuscript.

Another potential issue is that, in the field of nonequilibrium engines, the mere evaluation of efficiency may be not enough to assess the overall performance, see for instance Phys. Rev. Lett. 120, 190602 (2018). For instance, what is the effect of fluctuations in the power output?

This is indeed a very important and interesting question which we would like to address in the future as we mention in the “Discussion”, but it lays beyond the scope of our current work. To clearly contrast the case of prethermal and thermal working medium, we consider quasi-static transformations following the appropriate GGE. Hence, strictly speaking, we work in a regime of vanishing power (similarly to any other quasi-static description of heat engines), but notice that often the effective timescales required for prethermalization are relatively fast, as we comment at line 87, and are far less stringent than quantum adiabaticity; see lines 106-110. Overall, we now emphasize these aspects in the introduction, in lines 86-110. Finite-time effects could be introduced on a very short time scale $t \ll t_{\text{prth}}$ projecting the system completely out of equilibrium and making any description in terms of GGEs invalid. Studying this regime would require a completely different methodology, arguably less general. Another possibility would be to consider the interpolation times between the prethermal and thermal regime $t_{\text{prth}} < t < t_{\text{th}}$ and to explore the transient between the thermal and prethermal regime. In this transient, the GGE is expected to smoothly evolve from the prethermal to the thermal ensembles (see, e.g., Phys. Rev. Lett. 115, 180601 2015), but the details of the evolution depend on the microscopic model and how the conserved charges are broken, losing the general appeal of our results. Therefore, to

keep the message simple and the discussion contained, we focus on the quasi-static regime only.

What about the entropy production?

We thank the Referee for this question, which indeed is beneficial to better connect our findings with previous –and somehow more conventional– studies in quantum engines. As we clarified above in the report and in the revised version of our manuscript, in the adiabatic strokes we follow quasi-static transformations: in conventional working medium operating at thermal equilibrium, this is associated with entropy conservation. It turns out that this result can be generalized to prethermal states described by GGEs. Therefore, during the adiabatic strokes, entropy remain constant regardless of the medium instantaneously being in a thermal or prethermal state. Therefore the observed difference in performances cannot be justified from entropy considerations. We now discuss the lack of entropy production at the end of the first section in Results, more specifically at lines 179-195.

On a minor note, below I list some points the Authors may want to address to further improve the quality of their presentation:

1- In the results section, when dealing with the difference in the heat exchanged during the isochore strokes, the Authors may want to expand the discussion to further clarify and/or report the full expression for the heat, from which the leading order expression could be worked out.

We thank the Referee for this remark, which was also raised by another reviewer. We agree that expanding on this point would enhance the clarity of our discussion; therefore, we now provide more details at lines 246-256 in the revised manuscript.

2- In Fig. 2, the Authors could comment on the low values of the relative efficiency in the case of the Ising chain, occurring around $h=1$. Is that related to the criticality of the quantum Ising model?

The Referee here shared a very keen observation, which is, indeed, correct. We feel like including this discussion in the manuscript would divert the reader’s focus from our main scope, and therefore, we do not emphasize this aspect, but we provide here a short comment for the convenience of the Referee.

Near criticality and at low energy, one expands the energy in the Ising chain as $e(\lambda) \simeq 2\sqrt{(1-h)^2 + \lambda^2}$. If one uses this approximation to compute the energy-energy susceptibility $\langle H\partial_h H \rangle$, it vanishes. Therefore, according to the flow equations (2) in our manuscript, at criticality, the temperature during the adiabatic flow does not change $\partial_h\beta = 0$. Similarly, one can now consider the thermal mode density $\rho(\lambda) = \frac{1}{2\pi} \frac{1}{e^{\beta e(\lambda)} + 1}$: taking the derivative w.r.t. h , and using that $\partial_h\beta = 0$, one obtains that $\partial_h\rho = 0$ during the adiabatic stroke following the thermal equilibrium. However, in the Ising chain, the prethermal strokes are governed by Eq. (9) of our manuscript, with the additional input that F^{eff} vanishes (see lines below Eq. 9). Therefore, also in the prethermal strokes $\partial_h\rho(\lambda) = 0$.

In summary, at criticality, the system is unable to distinguish whether it is exploring a thermal or prethermal phase, and the associated efficiencies coincide.

3- In Fig. 3, the insets are not fully clear. Is the black (orange) curve related to prethermal (thermal) cycle?

Thanks for raising this question. The insets quantified the GGE's distance from equilibrium, measured as the relative distance in the L_2 norm between $\rho(\lambda)$ describing the GGE and $\bar{\rho}_{\text{th}}(\lambda)$ of a thermal state with the same energy (and magnetization for the XXZ model) as the GGE (i.e. the thermal state one would relax to from the GGE without further work injection). However, also motivated by the Referee's question, we realize the insets made the Fig. 3 too dense, while they are not critical to the main message. Therefore, we removed the insets in the resubmitted manuscript, which are now plotted and extensively discussed in the Supplementary Information.

4- In the Methods section, when reporting on the hydrodynamic projections' techniques, some readers may miss the reason why the projector onto the N' conserved charges is written in terms of the Q_j s rather than the deformed charges.

We agree with the Referee that this point deserves further clarification. While the perturbation is infinitesimal and thus sufficient to break the exact conservation of certain charges, it does not significantly deform their spectrum at leading order. This allows us to treat the thermal conserved quantities as forming a strict subspace of the prethermal ones. We clarify this assumption explicitly both at the beginning of the Result section (lines 144-153) and in the Hydrodynamic Projections paragraph of the Methods section.

Response to Referee 2

Summary: The manuscript presented by the authors brings forth the interesting idea of using prethermal substances for heat engines instead of thermal ones. They start by briefly presenting the concept of prethermality, and then proceed to apply it to Otto cycles. For which they study the ratio between the prethermal efficiency and the thermal one. Finally, they showcase their results for two different integrable systems.

Assessment: I find that the topic tackled by the authors is very original and genuinely very interesting to the broad community. However, it is my assessment that there is a large lack of clarity in the way they communicate their results. Given the scope of the journal, it would be essential that the authors make sure that their results are clearly presented for a wide audience of physicists. In the current form it is hard to understand even for someone from the same field. To the point that I cannot make a final assessment on the manuscript (see detailed analysis below). If I understood the submission correctly, I think the authors can change the way that the paper is presented so that I will be convinced to accept it. I therefore recommend for re-submission in Nature Communications after major revisions.

We thank the Referee for acknowledging the interest of our proposal and prompting us to revise the manuscript to increase the clarity of our presentation. We have carried out a major revision of the manuscript to this end and hope that the referee finds the revised manuscript fitting the standards of Nature Communications.

Comments and suggested changes: In this section I give a more detailed point-by-point assessment of the manuscript, some points are more important than others. Therefore I would like to put particular emphasis on points 2, 4, 5, 6, 7, 9, 11, 13, 15, 16, 18, and 21; as I think these points particularly need to be addressed before publication.

In what follows, we address each of the comments and suggested changes. For the sake of clarity, we color in cyan the issues that the Referee emphasizes most.

1. *Introduction* Since the requirement for a charge Q to be conserved is $[H, Q] = 0$, the space of conserved charges can only change in dimensionality if the perturbation ϵV changes the degeneracies of the Hamiltonian H . Though the authors seem to be claiming that in general the number of conserved charges is lower in general, from the argument I present above it could be either higher or lower. Also, how does one choose which are the charges that are relevant for the GGE?

We thank the Referee for this comment. While it is true that the space of conserved charges can, in principle, increase or decrease depending on the structure of the Hamiltonian, in practice nontrivial conserved quantities beyond energy, momentum, or particle number are typically present only in fine-tuned models. These are usually broken by generic perturbations, even when small. For example, in the Ising chain studied here, adding a longitudinal magnetic field breaks integrability and removes the conserved charges except the energy itself. In the XXZ chain, for example, the second-neighbor $S^z - S^z$ coupling breaks integrability, maintaining only the energy and z -magnetization as conserved quantities. Thus, in most physical scenarios, the number of conserved quantities is reduced when a perturbation is introduced. This is a fairly common scenario in several physical systems, as we now

emphasize in the introduction at lines 74-84.

Although it is a less natural point of view, one could swap the role of the two Hamiltonians and regard H_0 as a perturbation of $H_0 + \epsilon V$, obtained by removing the extra term. To avoid this confusion, we clarified our assumption in the definition of the perturbation at lines 74-77.

As for the charges relevant to the GGE, a widely accepted and safe statement is that in extended many-body systems all local (and, when relevant, quasi-local) conserved charges contribute, as already proposed in the seminal Ref. [39] Phys. Rev. Lett. 98, 050405 (2007), and further clarified in the review on integrable dynamics in Ref. [33] Journal of Statistical Mechanics: Theory and Experiment 2016, 064001 (2016), although the phenomenology is more general. We now mention both these references when first introducing the GGE at line 97, but we prefer not to provide excessive details on this interesting but very technical topic that would divert the focus from our core message.

2. Introduction It is clear why the authors present the concept of perturbed charges. However, it is unclear how introducing a perturbation and the symmetry breaking tie with the rest of the manuscript and their results.

We extensively revised the introduction to better explain the idea of prethermalization, which is an essential concept for our work. In this report, we take advantage to provide further clarifications. We seek to understand the role of prethermalization in changing the performance of a system. To this end, we need to compare engines working with comparable quantum matter; otherwise, other nuances of the systems could affect the result. To this end, we need to use the same thermal baths and two working media having the same Hamiltonian, up to small corrections that are inessential in computing the energy, but that when acting for a sufficiently long time can drive the system from a class of states –prethermal– to another –thermal–. In this case, which is the one addressed in our work, it is pertinent to compare two engines working in the two phases. We hope that the revised version of the introduction clarifies this important aspect.

3. Figure 1 It is unclear what is the space being represented (doing it in 3D does not help). And it is unclear what the wide arrows are supposed to represent.

Thanks for helping us to improve the clarity of the presentation with this comment. Figure 1 is meant to visualize the difference between a Otto cycle with thermalizing medium, whose state always moves within the “Thermal plane” described by fewer conserved charges (like the energy) and the control parameter χ only, and prethermal working medium where more information contained in a larger set of charges is needed to describe the adiabatic strokes. The figure is meant to visualize that the thermalizing medium explores a subset of the possible ensembles explored by the prethermalizing medium. We have updated the figure and the caption to better convey this message and improve the clarity overall.

In particular, we improved the connection between the figure and the main text. The previous label *Non-thermal space* is now referred to as *Prethermal phase*, consistent with lines 43, 60–61, 67, and 93-95. Similarly, the two wide arrows, representing heat exchange along the isochoric strokes, have been updated: we removed the labels *Heat absorbed* and *Heat released* and instead added Q_1 and Q_2 inside the arrows, in agreement with lines 65 and 68. Finally, we have completely rephrased the caption to highlight the importance

of the 3D plot: While the thermal cycle lies on the 2D thermal plane, the strokes in the prethermal cycle are allowed to leave the thermal plane and explore prethermal phases. To make this clearer, we also labeled the plane as *Thermal plane*.

4. Above eq. (1) “compare the case where the perturbation is active (thermal cycle) or not (prethermal cycle)” It is unclear how the presence of the perturbation changes anything in regards to why the cycle becomes prethermal or not. Since the system is connected to a thermal bath it will thermalize during that part of the cycle (open dynamics), it seems unclear how the prethermality can play a role in the designed cycles.

We thank the Referee for this insightful comment. We have expanded the discussion to clarify this point. As correctly noted, when the system is coupled to a thermal bath, the open dynamics drive the system to a thermal state. Indeed, the isochore parts of our cycle are completely standard.

However, during the adiabatic strokes, the working medium is isolated (unitary dynamics): We note, as discussed in the manuscript, that isolated quantum many-body systems are routinely realized in many experiments, for example, in cold atoms. In this case, the dynamics is unitary and, in relation to prethermalization, the system relaxes under its own dynamics to a proper GGE, which –in the regime of sufficiently slow changes of the control parameter– evolves in time retaining its GGE form. The timescale at which the transformation is performed compared to the thermalization timescales decides whether one operates in the prethermal or thermal regime. We significantly improved the introduction in the revised manuscript to better elucidate this point. In particular, at lines 85-110

5. Eq. (1) *How is the heat computed in this setting?*

During the adiabatic strokes of the cycles, the system is isolated: therefore, changes in the internal energy are due to work, without heat exchange. Instead, in the isochore transformation, the control parameter is kept constant: no work is exerted, and only heat is exchanged. Therefore, by tracking the internal energy of the system, we can compute both the work and the exchanged heat. We clarify these points in the description of the cycle, lines 58-68.

6. Below eq. (1) “Thermalizing medium is more efficient for a positive bath temperature, whereas prethermalization is convenient at a negative temperature” The authors should clarify what it means for a system to thermalize at negative temperature. It might be intuitive what this means for small systems where population inversion makes sense, but for large systems in the thermodynamic limit their Hamiltonian is unbounded from above, and therefore negative temperature seems to make little sense. Furthermore, how can this be implemented in a cycle? How does one take a macroscopic bath at negative temperature?

We thank the Referee for pointing out this aspect, which we realize was not properly clarified in the previous version of the manuscript. The Referee is right when saying that negative temperature is not possible if the Hamiltonian is unbounded from above. However, lattice systems have bounded kinetic energies and thus can realize negative temperature states. This is now commonly accepted in the field of many-body quantum systems and beautiful experiments have been realized; see Refs. [21,46,47] of the revised manuscript (which were already mentioned in the previous version as well). We improved our comment

on physical realizations of negative temperatures at the end of the introduction, and added further clarifications in the last section regarding the prospects of realizing our protocol in quantum simulators.

7. Results – The adiabatic flow equations. It is not clear to me why this result is mentioned here, as it seems to be uncorrelated to the results on the heat engine. Furthermore why do the generalized temperatures depend on the parameter χ ? Usually it should be the Hamiltonian that depends on that.

We thank the Referee for this question, which helped us in clarifying the setup –see in particular the introduction– in the revised version of the manuscript.

Our Otto cycle features four strokes: two isocores when the system is put in contact with the baths, and two adiabatic strokes. During the adiabatic strokes, the control parameter slowly changes, and the system follows an instantaneous GGE. To compute the work done, it is crucial to know the endpoints of the adiabatic stroke: In more standard scenarios with thermal phases where the Hamiltonian is the only conserved charge, entropy conservation is sufficient to identify the endpoints of the adiabatic strokes. However, if GGEs are allowed, entropy conservation is no longer enough (see the discussion at lines 179-196 in the revised manuscript) and one has to follow the entire evolution during the adiabatic stroke: the flow equations govern such an evolution. Indeed, the key point is that at the end of the adiabatic strokes (but before performing the subsequent isocore transformation), by evolving with different flow equations (thermal/prethermal) the system has reached different (pre)thermal final endpoints with different internal energies, finally affecting the work done and the heat exchanged in the next isocore transformation. Altogether, this is the mechanism that allows the two efficiencies to be different.

The Referee’s question about the χ –dependence of the generalized inverse temperatures should be now clarified: whereas the temperatures of the two baths are fixed (we improve our text to clarify this point), during the adiabatic strokes β_j evolve following the flow equations and are therefore functions of χ .

8. Below eq. (2) Why is there a c on the expected values? What does it represent?

Thank you. We use the common notation $\langle \dots \rangle_c$ to denote the connected part of the correlation functions $\langle \mathcal{O}_1 \mathcal{O}_2 \rangle_c \equiv \langle \mathcal{O}_1 \mathcal{O}_2 \rangle - \langle \mathcal{O}_1 \rangle \langle \mathcal{O}_2 \rangle$, but we agree with the Referee that this notation is not universally used. Therefore, we added an explicit definition in the revised version of the manuscript.

9. Results – Universality of infinitesimal cycles Why does the bath depend on χ , this is very confusing. As it makes it quite unclear on what is the working system and what is the bath.

Thank you for highlighting this point. We rephrase that sentence, emphasizing that the baths are at temperatures $\vec{\beta}$ and $\vec{\beta} + \delta\vec{\beta}$, while the stroke goes from χ to $\chi + \delta\chi$, see lines 212-215 of the revised manuscript.

10. Results – Universality of infinitesimal cycles I am not sure that the authors provide anything in the text that justifies the title. I think they have a reason to call it this way, i would like them to explain more clearly in what sense the infinitesimal cycles are universal.

Thank you for this question, we have now improved the text accordingly to clarify this point.

Studying many-body quantum systems with strong interactions is very challenging. In this respect, the integrable systems we analyze (the Ising chain and XXZ model) are useful toy models that allow us to non-perturbatively study (pre)thermal cycles and show the efficiency boost claimed in our work. However, integrable systems are rather special, and, if our results were confined to this case only, one would wonder whether the efficiency boost is due to prethermalization itself or to some peculiarity of integrable models. Remarkably, for infinitesimal cycles, we can obtain analytical, rigorous results in prethermal cycles without relying on integrability and are thus of much wider applicability. In particular, this analysis does not rely on the microscopic details of the model, such as the form of interactions, the dimensionality, or even the nature or exact number of the conservation laws. In this sense, the results shown in that section are *universal*.

To further clarify this point, we made the following changes: *i*) in the introduction at lines 120-124 we explain why the generality of the results in infinitesimal cycles (a similar explanation was already present in the previous introduction), *ii*) we update the title of the section to “Universal efficiency boost in infinitesimal cycles” to better emphasize the universality of the boosted performance, as opposed to the cycle itself, and *iii*) we reiterate in which sense the results are meant to be universal at lines 205-211.

11. Above eq. (4) why do the inverse temperatures change during the protocol?

Note: Eq(4) in the previous version manuscript is now Eq (5) in the revised version.

The answer to this question and the improvement to the manuscript are discussed in our reply to point 7 of the report.

Since the adiabatic stroke is performed quasi-statically, the system remains in a (pre)thermal state during the process. This allows us to associate a set of generalized inverse temperatures at each instantaneous value of the control parameter χ . The evolution of these inverse temperatures along the stroke is governed by the flow equation Eq. (2).

12. Eq. (4) Why is the second order of the expansion kept? It is strange that it is considered as it is negligible compared to the first order.

Note: Eq(4) in the previous version manuscript is now Eq (5) in the revised version.

It should be noticed that the first-order term in Eq. (5) is independent of the choice of the flow equations, and therefore it does not change if the thermal or prethermal phases are followed. To resolve the difference, one should go to the second order: we make this explicit at lines 224-228.

We note that the difference between the two strokes is expected to be at second order. This is evident from Eq. $\Delta H = \int_{\chi}^{\chi+\delta\chi} d\chi' \langle \partial_{\chi} H \rangle_{\chi', \vec{\beta}(\chi')}$ (above Eq. (5)): the first order is computed approximating $\langle \partial_{\chi} H \rangle_{\chi', \vec{\beta}(\chi')}$ to its value in the thermal bath at the beginning of the stroke, which is invariant whether the forthcoming stroke would be thermal or prethermal.

Therefore, differences arise at the next order.

13. Below eq. (4) “Combining Eq. (2) [...] the work difference $W_{pth}-W_{th}$ ” Why is this quantity being studied? How can these quantities be even compared in a fair manner? Furthermore, from equation (5) it seems that this difference is second order in $\delta\chi$, but then it is negligible to the actual amount of work being extracted. It is not very clear either how does the

Note: Eq(4) in the previous version manuscript is now Eq (5) in the revised version.

We regret that part of the Referee’s comment was cut in the PDF we received, and would be happy to address any further points should be shared.

We would like to first emphasize that work in both cycles (thermal and prethermal) is defined from the change of internal energy between the two strokes (as they do not exchange heat, whereas the isocore strokes exchange heat but not work). Therefore, the two quantities can be fairly compared. As we clarify in the Introduction (lines 52-56), both cycles have the same external baths, and (lines 149-153) we work in the regime where the perturbation ϵ governing prethermalization is small, in such a way both the thermal and prethermal Hamiltonians are comparable. These careful requirements ensure that, indeed, we can compare the prethermal and thermal cycles in a fair manner. Ultimately, we manage to express the relative efficiency in terms of the work difference δW (see line 258), which is then a convenient quantity to be addressed.

As a second observation, which is now clarified in our answer to point 12 of this report, the difference in work between the two cycles occurs at the second order in $\delta\chi$. Therefore, we agree with the Referee that the work difference is subleading to the work extracted, and this ultimately results in the fact that the difference in efficiencies is first order $\eta^{th} - \eta^{pth} \simeq \delta\chi$ (see also point 18). In summary, while studying infinitesimal cycles enables a clear and rigorous determination of the sign of the difference of the relative efficiencies, the result retains its infinitesimal nature. This also motivates our study of finite cycles (and thus, finite efficiency differences) in integrable models.

14. Eq. (5) I am not sure that introducing v_H for just this equation is very useful since it is non-zero in only one entry.

Note: Eq (5) in the previous version manuscript is now Eq (6) in the revised version.

We welcome the Referee’s suggestion and improve the notation accordingly.

15. Below eq. (5) “The heat absorbed in the thermal and prethermal cycles differs by the work difference at the end of the stroke before the isochore heating, which at the leading order for infinitesimal cycles is half of the total work difference $Q_{th} = Q_{pth} = \frac{1}{2}\delta W$ ”. It is completely unclear how this is true, as the authors did not provide any proof of this.

Note: Eq (5) in the previous version manuscript is now Eq (6) in the revised version.

We thank the Referee for this comment, also raised by another Referee. The identity follows

from Eq. (5) of the revised manuscript and basic energetic considerations, but we agree that it could be unclear to the reader. We now clarify this identity in the revised version of the manuscript, in lines 246-256.

16. Below eq. (5) Why are the authors only considering the relative efficiency? (they could write it similarly to the previous equations at this point, but they don't) There are many other typical quantities one could check beyond this. First, how does the efficiency of the prethermal engine compare to Carnot efficiency? Can it beat it since we are out of equilibrium? Is there a similar bound one can find for it? Second, what about power? (And efficiency at maximum power?) Third, what about power fluctuations?

Note: Eq (5) in the previous version manuscript is now Eq (6) in the revised version.

We thank the Referee for these insightful comments, which echo similar points raised by Referee 1. As we address the same questions about power and power fluctuations, we would first like to share the same answer we gave to Referee 1, before touching on Carnot efficiency.

About power and power fluctuations:

This is indeed a very important and interesting question which we would like to address in the future as we mention in the “Discussion”, but it lays beyond the scope of our current work. To clearly contrast the case of prethermal and thermal working medium, we consider quasi-static transformations following the appropriate GGE. Hence, strictly speaking, we work in a regime of vanishing power (similarly to any other quasi-static description of heat engines), but notice that often the effective timescales required for prethermalization are relatively fast, as we comment at line 87, and are far less stringent than quantum adiabaticity; see lines 102-110. Overall, we now emphasize these aspects in the introduction. Finite-time effects could be introduced on a very short time scale $t \ll t_{\text{prth}}$ projecting the system completely out of equilibrium and making any description in terms of GGEs invalid. Studying this regime would require a completely different methodology, arguably less general. Another possibility would be to consider the interpolation times between the prethermal and thermal regime $t_{\text{prth}} < t < t_{\text{th}}$ and to explore the transient between the thermal and prethermal regime. In this transient, the GGE is expected to smoothly evolve from the prethermal to the thermal ensembles (see, e.g., Phys. Rev. Lett. 115, 180601 2015), but the details of the evolution depend on the microscopic model and how the conserved charges are broken, losing the general appeal of our results. Therefore, to keep the message simple and the discussion contained, we focus on the quasi-static regime only.

About Carnot efficiency:

We notice that the Carnot efficiency, as an upper bound to the engine’s efficiency, is well defined only for positive temperatures. In this regime, we show that thermal working medium (whose efficiency is bounded by the Carnot inequality) is more efficient than prethermal medium, which therefore cannot beat Carnot. Instead, in the case of thermal baths with negative temperature, the efficiency of Carnot cycle at negative temperatures is a delicate issue; see, e.g., Refs. [14] and [47] in the revised manuscript. In summary, we feel like a

direct comparison with Carnot efficiency is not meaningful in our setup, and on the contrary, could divert the reader’s attention from the main focus, that is the relative efficiency of engines operating with thermal and prethermal working medium, and not their efficiency with respect to the Carnot bound.

We expanded the discussion about the computation of work and heat, in such a way that it is immediate for the reader to recover the expression for the efficiency, which is reported in SI for the specific cases of the Ising and XXZ chains we analyze. Since the expression for the efficiency can be almost immediately recovered from Eq. (6), but it is somewhat lengthier, we feel it is unnecessary to report the explicit equation in the manuscript. The discussion we provide is sufficient to convey the core message of our work on the relative efficiency between thermal and prethermal working medium, see the expression for $\eta^{\text{pth}}/\eta^{\text{th}}$ at line 258 and the related discussion.

17. Results – Prethermal finite cycles in integrable models. This section seems to be very nicely explained, but it is difficult to understand the presented application of the results of the previous sections because it is not exactly clear what we are looking at.

We thank the Referee for the positive assessment of this section. We agree that the connection to the previous results may not have been sufficiently clear in the original version. We hope that this link appears more explicitly in the revised manuscript. In the section “Universal efficiency boost in infinitesimal cycles”, we analytically demonstrate that a prethermal working medium can enhance the efficiency of an Otto cycle operated with a negative-temperature reservoir in the limit of infinitesimal cycles, where the efficiency gain is also infinitesimal. The goal of the section “Prethermal finite cycles in integrable models” is to provide a workable example where finite cycles can be studied and explore whether the efficiency enhancement persists for finite cycles. We exploit hybrid analytical-numerical techniques from integrable systems, allowing us to go beyond perturbative regimes and explore finite-temperature thermal phases and far-from-equilibrium nonthermal ones. This has been made possible by the recent theoretical advances in Generalized Hydrodynamics. The results presented in this section confirm that the efficiency boost is not limited to the infinitesimal regime. In the Supplementary Information we expand the discussion, remarking that we carried a systematic investigation in finite cycles, always finding consistent agreement with the analytical predictions we obtained in infinitesimal cycles.

18. Fig. 2 It is very difficult to interpret these plots, and the captions (or text) do not really help the reader to understand what do the graphs tell us. Furthermore the normalization chosen adds a level of encoding that makes it even less understandable. For example, on the green plots: why are we dividing the efficiency by δh ? People are used to efficiency to be bounded by 1, but here you reader does not have any reference. I hope there is a good reason for why one would add the unusual normalization. (also, what are the units of this quantity?)

We thank the Referee for these comments, which helped us to clarify the revised caption of Fig. 2 in the resubmitted manuscript. Let us take the opportunity in this report to provide further clarifications.

First, we would like to stress once again that the main focus of our investigation is on the

engine’s efficiency and that is thus the quantity we focus on. In infinitesimal cycles, the efficiency ratio for cycles performed in the neighborhood of temperature β and control parameter χ is (see line 258) $\frac{\eta^{\text{pth}}}{\eta^{\text{th}}} = \left(1 + \frac{\delta\mathcal{W}}{\mathcal{W}^{\text{th}}}\right) \left(1 - \frac{\delta\mathcal{W}}{2Q_{\text{abs}}^{\text{th}}}\right)^{-1}$: While the sign of $\delta\mathcal{W}$ is already sufficient to determine whether $\frac{\eta^{\text{pth}}}{\eta^{\text{th}}}$ is greater or smaller than one, its actual value depends on $\delta\mathcal{W}$ in a nontrivial way, making difficult plotting this function of β and χ only. To ease the data visualization, as explained in the caption of Fig. 2, we focus on skewed infinitesimal cycles where the stroke is much smaller than the temperature gradient in such a way $\mathcal{W} \ll Q_{\text{abs}}$. In this regime, the efficiency is infinitesimal $\eta = \delta\chi \times (\text{function of } \beta \text{ and } \chi \text{ only})$, and the efficiency ratio becomes $\frac{\eta^{\text{pth}}}{\eta^{\text{th}}} - 1 = \frac{\delta\chi}{\delta\beta} \times (\text{function of } \beta \text{ and } \chi \text{ only})$: the normalization in Fig. 2 reflects this form, showing that part of the efficiency and of the efficiency ratio that does not depend on the stroke size $\delta\chi$ or the distance of the two baths $\delta\beta$. These points are now clarified in the caption of Fig. 2. Since the explicit calculations for the final formulas are easily obtained from Eq. (5) and (6) and do not add significant insight to the discussion, we do not report them in the main manuscript and instead discuss them in the Supplementary Information.

Lastly, also in connection with point 20 of this report, we would like to comment on the choice of units: throughout the manuscript, to ease the notation, we consistently use dimensionless units. This is now explicitly mentioned in lines 277-278.

19. Above eq. (8) “while other choices are associated with different GGEs” for example?

We acknowledge that this sentence in the previous version of the manuscript could have been misleading, and therefore we revised it in the resubmitted manuscript; see lines 295-298.

We would like to use this reply to further comment on this matter. Determining the connection between the quasimomenta distribution $\rho(\lambda)$ and GGEs has been the subject of intensive research in integrable models. While the problem of retrieving the root density given a certain GGE (comprising, as a simple example, thermal states) has been solved since long in integrable models within Thermodynamic Bethe Ansatz (see Methods and SI), the opposite implication has a more recent history. Given a certain $\rho(\lambda)$, is that a physical choice associated with a certain GGE? The answer is positive, and while this is relatively simple in non-interacting integrable models (see, e.g., Phys. Rev. Lett. 106, 227203 2011), in the interacting case, it has been settled only more recently through the discovery of quasilocal charges (e.g., see Phys. Rev. Lett. 115, 157201 2015).

While interesting, deepening this direction would divert the reader’s focus in excess from our core message. Therefore, we limit ourselves to point out this connection and refer the reader to the nice review [33] (Journal of Statistical Mechanics: Theory and Experiment, 064001 (2016)), where this issue is carefully addressed.

20. Fig. 3 Some numbers are being chosen to compute the plots, but no units are provided. I assume some constants have been set to 1 somewhere.

The Referee is right, we opt to use adimensional units for the sake of notation, but we did not explicitly state it in the previous version of our manuscript. This choice is now made clear in lines 277-278.

21. Generally on the manuscript. I usually am against commenting on the presentation style of a manuscript, as it should be a free choice of the authors (within the limits of where they decided to publish their results). However, I believe that here the presentation style has strong negative effects on the communication of the science done by the authors, and is therefore relevant to the discussion. Here the authors chose the quite dry format 1) introduction 2) result A 3) result B ... N - 1) conclusion N) methods. I don't think that this is helping in any way the communication of the results. I believe it is actually impairing it, as some things seem very out of context or not introduced at all throughout the manuscript. Very often it is also not clear why the authors study some quantities rather than others. I would like the authors to consider taking a more usual approach where they properly and gradually introduce what is needed to understand their results (including the derivation of the results, as they don't seem very long by looking at the Methods section).

We thank the Referee for spurring us in greatly enhancing the clarity of our manuscript through the many suggestions presented in their report. We hope that the several improvements we made, together with the further clarifications provided in the report, make the communication of our results clear and precise, and overall accessible and appealing for the broad audience that Nature Communications aims for.

Nonetheless, we respectfully believe there is no need for a complete overhaul of the manuscript's organization, which has been carefully thought when preparing our submission. We would like to quickly review the manuscript's structure and our reasoning.

- Introduction. We carefully plan it to give the reader a general overview of quantum engines, and in particular, of the opportunities offered by quantum many-body systems. We introduce prethermalization, state our main result, and clarify to the reader less familiar with negative temperatures that such states are possible, announcing that we envision how our results could be concretely observed in the state-of-the-art quantum simulators. By the end of the introduction, the reader is informed of the scope of our work, the general framework, and the statement of the results, even before diving into the core of the manuscript.
- Results. General comment: We believe we present some very general results, which are largely independent of the fine details of the model. Furthermore, these results are based on some very general thermodynamic inequalities, which can be understood by the reader without a deep background in prethermalization mechanisms, such as integrability. We agree with the Referee that, in other instances, building the storyline on simple examples before touching upon the more general results is an effective communication strategy. However, in our case, the concrete example of a prethermalizing model is more technical than the general analysis we provide on infinitesimal cycles themselves. Indeed, since prethermalization belongs to many-body systems, we cannot provide a simple few-level example. Next in complexity, we find integrable models: their introduction needs some background, as the Referee

can see in our manuscript. Furthermore, presenting them at the beginning of our Results section may leave the reader with the idea that our results are valid only in these special models, hampering the understanding of the generality of our findings. The structure we gave to the Result section stems from these considerations. More specifically:

- The adiabatic flow equations. The flow equations are the technical backbone of our work, describing the system’s evolution during the adiabatic strokes. Furthermore, in the revised version of the manuscript, we further clarify the connection with the conventional notion of adiabaticity based on entropy conservation. This section gives the reader the technical tools to understand the rest of the manuscript.
 - Universal efficiency boost in infinitesimal cycles. This section, albeit limited to infinitesimal cycles, embraces the full generality of our results, showing the validity of our claim regardless of the details of the model, but only requiring the breaking of certain conservation laws. In this sense, the results are universal (see also point 10 of this Report). Furthermore, the derivation is remarkably simple on the technical side, as it involves a simple Taylor expansion, plus recognizing a scalar product within the hydrodynamic projections’ approach. In the revised version of the manuscript, this section has been further improved and clarified, leaving only some lengthier technical parts to the Methods (e.g., the hydrodynamic projections), while still explaining all the main points within the section.
 - Prethermal finite cycles in integrable models. Albeit general, the results in the previous section pertain to infinitesimal cycles, therefore it is important to see whether they remain valid for finite cycles. For doing this, we need to provide an example of many-body quantum systems featuring prethermalization, and such that they are amenable to quantitative analytical predictions at the many-body level: integrable models are the ideal venue for this goal. We present the generalities of this class of models, and the GHD approach (which, within integrable models, parallels the flow equation (2)). We then connect with the previous section by comparing the efficiency of infinitesimal cycles in our working example, and finally moving to finite cycles, further corroborating our results.
 - Quantum engine simulators. The previous sections pertain to a purely theoretical study, whereas this section provides a careful assessment of the relevance of our general results for the state-of-the-art experiments. While a detailed analysis of a given experiment is beyond the scope of our work, we indicate that the main ingredients for experimentally benchmarking our findings are already present in quantum simulators and discuss the general strategy, from realizing negative temperatures to controlling prethermalization and probing the work done in the adiabatic strokes.
- Discussion. We summarize our results and discuss how they are seminal to interesting future directions, some of which are also mentioned by the Referees, such as studying finite-time protocols and the related questions of power output and fluctuations.

- **Methods.** We present in a compact and clear manner the technical aspects of our work. We report technicalities here to not hamper the flow of the main text, and provide, at the same time, all the tools to the reader to reproduce our results. While the manuscript is self-contained even in its more technical aspects, we nonetheless provide a further, even lengthier and more pedagogical discussion in the Supplementary Information as a further backup for the reader.

In summary, we believe our manuscript is well-structured as it is. We are grateful to the Referee for the many suggestions that helped us in improving the presentation in each of its parts, and are confident of the clarity of the revised manuscript, without the need for additional restructuring. We hope that the referee shares this view, in agreement with the positive feedback we have received from other readers (those listed in the acknowledgements, among other colleagues).

Response to Referee 3

The authors show that prethermal quantum heat engines can be more efficient than thermal ones, for negative temperatures, while the opposite is true for positive temperatures. The authors start with a general formalism, and then consider specific examples of two integrable models. The work is timely, rigorous, and the results seem important, and are applicable to a broad category of systems.

We thank the Referee for the overall positive assessment of our work and for recognizing its importance, rigor, and broad applicability.

*My main comment is, the authors claim that advantage in efficiency exists for negative temperatures. However, negative temperatures are associated with ergotropy, and there already has been many works which show that negative temperatures enhance the performance of quantum engines. For example, see *Phys. Rev. Lett.* 112, 030602 (2014) and *PRX Quantum* 3, 040334 (2022). So given that it is already well known that negative temperatures boost the performance, I suggest the authors add a discussion on why they claim their results show something very novel. Does ergotropy play any role here?*

We thank the Referee for prompting us to clarify the novelty of our findings and the role of ergotropy, and suggest relevant references that are now properly cited in our revised manuscript. Before discussing ergotropy in further detail, let us provide a general clarification. In investigating the advantages of negative temperatures in an engine’s performance, one should always specify “with respect to what”. In the studies pointed out by the Referee—as well as in other works in the literature on this aspect—the performance of a certain working medium at positive temperature is contrasted with the same working medium, but at negative temperatures. In short, the comparison concerns the baths rather than the working medium. Our case is substantially different, as while considering the same baths we are interested in performance differences of two working media that differ in their ability to sustain prethermal phases or not. Hence, our goal and result substantially differ from those already present in the literature and, to the best of our knowledge, provide an entirely novel perspective at the interface of engines and many-body quantum dynamics.

Let us now provide a more detailed discussion. We note that the work *Phys. Rev. Lett.* 112, 030602 (2014) concerns a QHE with a single-trapped ion as a working medium. As for *PRX Quantum* 3, 040334 (2022), it relies on a 2-, 3-, and up to 7-level description of individual Cs atoms. It is fair to say that the community of quantum thermodynamics has largely ignored many-body quantum effects for several decades; see Introduction for a more detailed exposition (Widera’s works offer a nice exception in this regard). Our work exploits the conservation laws inherent to many-body quantum systems, which give rise to the notion of prethermalization. In this sense, our findings cannot be derived or understood in terms of previous works relying on a single-particle description, even if a common feature is the use of negative temperatures. The need to transcend single-particle analysis and embrace many-body effects is central to our manuscript, as already emphasized in the Introduction, and presents an opportunity that arises in scaling QHEs. Our work harnesses prethermalization to enhance the performance of the QHE when operating at negative temperatures. We do

not think the concept of ergotropy provides the right means to interpret our results for two reasons. The first reason stems from the definition of ergotropy itself, as the maximum work that can be extracted from a quantum state with arbitrary unitary operations. Many-body quantum systems are extremely complex, and only certain unitary operations are physical –(pre)thermalizing dynamics is one of these– and thus it is not relevant to maximize on all possible unitary choices. The second, is that we are not comparing the maximum work that can be extracted from two different states, which would be quantified by different ergotropies, but rather the work –or better, the efficiency– that can be extracted from the same initial state –ingenerated by the contact with the initial bath– that undergoes two different evolutions, one exploring thermal phases and the other prethermal phases.

In summary, we provide new insight into the relative performance of quantum *many-body* engines by using the appropriate tools for (non)equilibrium quantum physics. Although our results may not be cast in simple terms using conventional concepts such as ergotropy, they are grounded in an exact analysis and are relevant for quantum simulator experiments.

Also given that negative temperature is the key here, I would suggest adding that in the title as well. But I leave this to the consideration of the authors.

Thanks for this suggestion, which we have followed.

My other minor comments are:

1. In the description of strokes, in page 1, above Fig. 1, the authors write that during the adiabatic stroke the system follows a thermal or prethermal state. However, when a Hamiltonian parameter is changed at a finite rate, in general the system goes out thermal equilibrium. So do the authors consider slowly changing Hamiltonian, or some control techniques, such as shortcuts to adiabaticity?

We thank the Referee for this comment, which motivated us to largely improve the introduction to better clarify these aspects. Let us give a short answer before providing more details: we assume no external controls here, and the relaxation to the (pre)thermal state is completely driven by the many-body dynamics.

Let us provide further clarifications. The topic of (pre)thermalization in *isolated* quantum many-body systems has been intensively studied in the last decades. Given an initial state, pure or thermal, and unitary evolving according to some time-independent Hamiltonian, under what conditions, and in which sense, thermalizes? Under mild assumptions on the initial state and on the Hamiltonian, mostly its locality and the thermodynamic limit, it is now understood that information gets dissipated in highly non-local degrees of freedom, and the state shows relaxation to a stationary state in a local sense. Namely, from the viewpoint of any correlation function of all local observables, at large time the state is indistinguishable from an appropriate mixed state determined by the appropriate statistical mechanical description embodying the symmetries and conservation laws of the dynamics. In the cases we focus on, this stationary state is the GGE. These concepts are nicely discussed in the seminal work introducing the GGE (Ref. [39], Phys. Rev. Lett. 98, 050405 (2007)), and in the Special Issue on dynamics in quantum integrable models (Ref. [33], Journal of Statistical Mechanics: Theory and Experiment 2016, 064001 (2016)). A further step is including prethermalization as a gentle crossover between an apparent relaxation to a

state with some given conservation laws, to another one with a reduced set of symmetries: we now better introduce these concepts in the introduction (lines 85-110), quoting the appropriate literature. During the adiabatic strokes, our system is isolated and therefore it tries to reach the appropriate (pre)thermal state, while the control parameter is changed in time. If the change is sufficiently slow, the system is approximately described by a time-dependent GGE. We further clarify these aspects in the introduction (lines 102-110). Note that the required timescales are substantially smaller than those of quantum adiabaticity, as discussed in the manuscript.

2. Similar comment for the isochoric stroke as well - the authors write that the system remains thermal. Do they mean the system remains at thermal equilibrium with the bath at the end of the stroke, or does the system remain at thermal equilibrium at all times, even during the stroke? I would expect the former, since I understand that the system is coupled to a bath at a constant temperature, and the heat flow brings the system to thermal equilibrium at the end of the stroke. The authors should clarify this point.

We thank the Referee for this helpful observation. We have revised the description of the isochoric stroke to remove the potentially misleading comment about the system remaining thermal throughout. As correctly pointed out, during the stroke, the system is coupled to a thermal bath at constant temperature and relaxes toward thermal equilibrium. What matters for our analysis is that the system reaches thermal equilibrium with the bath at the end of the stroke. The specific relaxation dynamics during the stroke do not affect our efficiency results.

3. The authors consider N conserved quantities. Is N of the order of system size, or is it of the order of the Hilbert space dimension of the system?

As our results are framed in the thermodynamic limit, the Hilbert space dimension is divergent. For example, in the transverse field Ising model, the Hilbert space dimension diverges exponentially as 2^L as the system size $L \rightarrow \infty$. The number of conserved quantities, denoted N , is introduced to keep the discussion general and is not tied to the system size. We notice that integrable systems feature infinitely many conserved charges in the thermodynamic limit, as their number diverges extensively in the system's size $N \sim L$ (yet, their number diverges much slower than the Hilbert space dimension). Even in this case, the GGE remains a well-defined concept as discussed, for example, in Ref. [33] *Journal of Statistical Mechanics: Theory and Experiment* 2016, 064001 (2016), or in the founding papers of Generalized Hydrodynamics reported in our manuscript.

We would like to emphasize once more that our results are expected to be universal and not restricted to integrable models only, as we show in the analysis of infinitesimal cycles. In this case, we obtain analytical and rigorous results with no assumptions on the number of conservation laws, the form of the dynamics, or the dimensionality of the system.

Response to Referees for resubmission of NCOMMS-25-28247

Below, we provide a detailed point-by-point answer to the questions and suggestions raised by the Referees, reporting quotes from their reports in *italic*, whereas our answers are reported in boxes.

Response to Referee 1

I was asked to reconsider the manuscript NCOMMS-25-28247A-Z, entitled “Universal efficiency boost in prethermal quantum heat engines at negative temperature”. Compared to its previous version, the title of the manuscript has been changed and a more accurate explanation of prethermal against thermal cycles has been reported. The manuscript has also benefited from the inclusion of a paragraph related to entropy production. Moreover, the main results have been emphasized, and additional details have been provided in the supplemental material. Although the Authors replied to all my points, I confirm that the novelty and the actual significance of the presented results to the field of quantum heat engines (QHE) is difficult to assess. Nevertheless, the manuscript could more likely be relevant to the field of quantum simulators.

We provide a detailed response below, which we hope brings further clarifications. We also anticipate this work to be of relevance across different fields due to its interdisciplinary character, merging recent developments in quantum many-body physics, ultracold atomic gases, and quantum thermodynamics. In particular, we believe our work has the merit of bridging between more canonical studies of QHE with the fast-growing quantum simulators’ community. Leading experimental groups are preparing to leap forward in this direction [see eg. PRX Quantum 2, 030310 (2021)] and, since quantum simulators are ideal platforms to study diverse venues for prethermalization, our work gives an exciting and concrete groundwork for future developments.

There is no doubt that prethermalization is a many-body effect with no single or few-body counterpart. However, the issue of whether the “[...] breakdown of conventional thermodynamics [...]” (i.e, the absence of thermalization) “[...] can be advantageous for a many-body quantum engine [...] ” has been already addressed in the past, focusing on the very special case of ergodicity breaking in the MBL phase, see for instance Phys. Rev. B 99, 024203 (2019). Indeed, as reported in recent review papers on the subject, QHEs operating with many-body systems as their working medium have been studied for more than a decade.

We thank the referee for this insightful comment, pointing out a preceding example in which the absence of conventional thermalization, has also been discussed and used in a quantum heat engine. We have thus cited it in the revision. We would like to emphasize that, due to the very different nature of prethermalization in MBL (not described by a GGE), the suggested reference has a complementary scope to our work. While disorder is a well-study source of non-ergodicity, we believe that the selective breaking of conservation laws or symmetries is an interesting scenario naturally realized in several experimental platforms. Furthermore, we would like to emphasize our work is the first combining quantum heat

engines with generalized hydrodynamics (GHD), giving quantitative analytic predictions on the one hand, and bringing our proposal closer to a factual realization since GHD has already been successfully used to describe several cold atoms experiments.

Moreover, as stated by the same Authors, when compared to previous proposals in the field, this engine delivers an amount of power that is close to zero.

Adiabatic thermodynamic cycles and heat engines are at the foundation of both classical and quantum thermodynamics. With or without prethermalization are expected to have vanishing output power. Of course, this does not challenge their utility, both as a theoretical reference framework, and in practical heat engines, given that adiabaticity does not necessarily involve infinite cycle times, and provides an efficient description in finite times too. Furthermore, we would like to emphasize we do not work in a regime of quantum adiabaticity (QA), where the system follows an instantaneous eigenstate, but in a regime of (pre)thermalization, where the system explores a well-defined ensemble of states (i.e. the GGE). This distinction is important both conceptually and practically, as in many-body interacting quantum systems the applicability of QA requires to compare the protocol's timescale with an energy gap, usually closing in the thermodynamic limit. Hence, the realization of QA may require prohibitively long time scale. In contrast, (pre)thermalization is a concept that is applied already in the thermodynamic limit, hence its time scale remains finite. As we discuss in our proposed testbed in quantum simulators, prethermalization scales usually requires only few hopping-times, well within experimental reach. The fact that GHD (which assumes instantaneous and local prethermalization) has been successfully used to describe cold atoms experiment further show the applicability of our findings to real-life (and obviously finite-time) experiments. The important distinction between QA and the prethermalization framework here considered is thoroughly detailed in the introduction. In the revised version, we further emphasize this aspect in the conclusive discussion as well.

While it is true that the thermodynamics of Generalized Gibbs Ensembles (GGE) has mainly been studied in the context of non-thermal baths, the prethermal state of the working medium, as stated by the same Authors, is modeled as a peculiar instance of GGE, which is adiabatically driven in time and sustains a number of conserved charges that can be deformed or broken due to the perturbation. Consequently, the occurrence of unconventional “efficiency boosts” due to the combined effect of nonequilibrium states of the working medium and of the baths cannot be considered as totally unexpected.

We thank the referee for this comment, which gives us a further chance to clarify the perspective of our work, its generality and impact. The possibility of taking advantage of additional degrees of freedom (i.e., the conserved charges) in the thermodynamics of the working medium is what motivated us in devising this project. Will these charges increase efficiency, or work against it? This was the question that originally drove us. In principle, playing with arbitrary and perhaps artificial GGEs, one could have argued (as the referee suggests) that one could improve the efficiency. However, here we take physics' constraints into account, considering only GGEs that are explored by changing a control parameter. Given this constraint, how do the extra conservation law impact the efficiency? We believe that the answer to this question is a-priori very non-trivial.

Without knowing the results reported in our work, one could have guessed both ways, or

maybe having imagined a case-by-case outcome. It came to our surprise that both scenarios are possible, namely extra conservation laws can be beneficial or not, and this is universally determined by the bath's temperatures. The simple, yet non-trivial and general answer we provide to the simple question motivating our work is a further strong motivation for the broad impact and interest of our result.

As already detailed in my first report, this manuscript stands out from previous works on many body QHEs mainly due to its rigorous proof of the properties of prethermal cycles. Indeed, the ingenious use of the perturbation-deformed charges provides a remarkable insight, i.e., adiabatically driven nonequilibrium prethermal states are more suitable to harness quantum resources (i.e., population-inverted negative temperature baths) when compared to thermalizing states, while the opposite holds true for more conventional resources, i.e., positive temperature baths. Due to the wise use of the powerful hydrodynamics formalism, the Authors prove this result to be universal, i.e., it can be extended to a wide class of interacting many-body systems. Eventually, the Authors also suggest several experimental platforms, based on ultracold atoms trapped in optical lattices, to probe their theoretical results.

We thank the referees for the positive assessment of our work.

Response to Referee 2

I want to thank the referees for their responses and for the modifications they implemented in the manuscript. I think they have properly addressed all of my concerns and that the current version of the manuscript is ready for publication.

We thank the referee for the positive assessment of our work and for recommending it for publication in Nature Communications. In particular, we are grateful to their previous report, which helped us in greatly improving the communication of our results. We are delighted Referee 2 is satisfied by the improvements, and by our revision.

Response to Referee 3

The authors have responded to the comments satisfactorily, and also modified the manuscript accordingly to highlight the main results and improve the clarity. In particular, the authors emphasize that the origin of the improvement in the performance for negative temperatures stems from the pre-thermal nature of the many-body quantum system. These results seem universal, and in my opinion can be a significant contribution in the topic of many-body quantum engines. However, preparing a bath at a negative temperature incurs a cost. Will these advantages change when that cost is taken into account?

I recommend publication of the manuscript once the authors have added a discussion on the above aspect.

We thank the referee for helping us to improve the clarity fo the presentation by pointing out that the preparation fo thermal reservoirs at negative temperatures generally involves some engineering. Indeed, such reservoirs are unlike the Maxwell-Boltzmann, Fermi-Dirac, Bose-Einstein and generalized Gibbs distribution generally found in nature “for free”. By

contrast, the preparation of negative temperature reservoirs require population inversion, which generally requires some transfer of energy and doing some work. While such preparation is platform dependent and there is no consensus on the actual cost of their preparation, they are widely appreciated to be useful in applications, as reflected by the cited references and our work. The rationale is that one can generally argue for scenarios in which the practical benefits of the boost (not necessarily given by an assessment of quantum resources) exceeds those of the cost of preparing the negative temperature reservoir. This is in line with previous concerns about proposal using them, e.g. in [11-15]. We have added a discussion in the text to reflect this aspect, under the section on "quantum engine simulators" as well as in the closing Discussion. Furthermore, we would like to emphasize that in quantum simulators engineering synthetic lattice Hamiltonians, which are arguably the most natural platforms to observe our findings, the state preparation usually does not achieve directly thermal states, but rather product states. Such a product state can, after thermalization, results in both a positive temperature or negative temperature Gibbs ensemble depending on the details of the implemented Hamiltonian (as we discuss in the quantum simulator's paragraph). Hence, at least in this case, we do not think the "cost of state preparation" is a well-defined question. We hope that with this revision the referee finds the manuscript suitable for publication in its current form.

Response to Referees for resubmission of NCOMMS-25-28247

Below, we provide the final reply to Referee 3.

Response to Referee 3

The authors have considered my comments and modified the manuscript accordingly. I recommend publication of the manuscript in the present form.

We sincerely thank Referee 3 for the positive feedback and for recommending our work for publication in Nature Communications. All reports were particularly helpful in guiding us to improve the clarity and presentation of our results.

Report on NCOMMS-25-28247: “Universal efficiency boost in prethermal quantum heat engines”

Summary The manuscript presented by the authors brings forth the interesting idea of using prethermal substances for heat engines instead of thermal ones. They start by briefly presenting the concept of prethermality, and then proceed to apply it to Otto cycles. For which they study the ratio between the prethermal efficiency and the thermal one. Finally, they showcase their results for two different integrable systems.

Assessment I find that the topic tackled by the authors is very original and genuinely very interesting to the broad community. However, it is my assessment that there is a large lack of clarity in the way they communicate their results. Given the scope of the journal, it would be essential that the authors make sure that their results are clearly presented for a wide audience of physicists. In the current form it is hard to understand even for someone from the same field. To the point that I cannot make a final assessment on the manuscript (see detailed analysis below). If I understood the submission correctly, I think the authors can change the way that the paper is presented so that I will be convinced to accept it.

I therefore recommend for re-submission in Nature Communications after major revisions.

Comments and suggested changes In this section I give a more detailed point-by-point assessment of the manuscript, some points are more important than others. Therefore I would like to put particular emphasis on points 2, 4, 5, 6, 7, 9, 11, 13, 15, 16, 18, and 21; as I think these points particularly need to be addressed before publication.

1. *Introduction* Since the requirement for a charge Q to be conserved is $[H, Q] = 0$, the space of conserved charges can only change in dimensionality if the perturbation ϵV changes the degeneracies of the Hamiltonian H . Though the authors seem to be claiming that in general the number of conserved charges is lower in general, from the argument I present above it could be either higher or lower. Also, how does one choose which are the charges that are relevant for the GGE?
2. *Introduction* It is clear why the authors present the concept of perturbed charges. However, it is unclear how introducing a perturbation and the symmetry breaking tie with the rest of the manuscript and their results.
3. *Figure 1* It is unclear what is the space being represented (doing it in 3D does not help). And it is unclear what the wide arrows are supposed to represent.
4. *Above eq. (1)* “compare the case where the perturbation is active (thermal cycle) or not (prethermal cycle)” It is unclear how the presence of the perturbation changes anything in regards to why the cycle becomes prethermal or not. Since the system is connected to a thermal bath it will thermalize during that part of the cycle (open dynamics), it seems unclear how the prethermality can play a role in the designed cycles.
5. *Eq. (1)* How is the heat computed in this setting?
6. *Below eq. (1)* “Thermalizing matter is more efficient for a positive bath temperature, whereas prethermalization is convenient at a negative temperature” The authors should clarify what it means for a system to thermalize at negative temperature. It might be intuitive what this means for small systems where population inversion makes sense, but for large systems in the thermodynamic limit their Hamiltonian is unbounded from above, and therefore negative temperature seems to make little sense. Furthermore, how can this be implemented in a cycle? How does one take a macroscopic bath at negative temperature?

-
7. *Results – The adiabatic flow equations* It is not clear to me why this result is mentioned here, as it seems to be uncorrelated to the results on the heat engine. Furthermore why do the generalized temperatures depend on the parameter χ ? Usually it should be the Hamiltonian that depends on that.
 8. *Below eq. (2)* Why is there a c on the expected values? What does it represent?
 9. *Results – Universality of infinitesimal cycles* Why does the bath depend on χ , this is very confusing. As it makes it quite unclear on what is the working system and what is the bath.
 10. *Results – Universality of infinitesimal cycles* I am not sure that the authors provide anything in the text that justifies the title. I think they have a reason to call it this way, I would like them to explain more clearly in what sense the infinitesimal cycles are universal.
 11. *Above eq. (4)* why do the inverse temperatures change during the protocol?
 12. *Eq. (4)* Why is the second order of the expansion kept? It is strange that it is considered as it is negligible compared to the first order.
 13. *Below eq. (4)* “Combining Eq. (2) [...] the work difference $\mathcal{W}^{pth} - \mathcal{W}^{th}$ ” Why is this quantity being studied? How can these quantities be even compared in a fair manner? Furthermore, from equation (5) it seems that this difference is second order in $\delta\chi$, but then it is negligible to the actual amount of work being extracted. It is not very clear either how does the
 14. *Eq. (5)* I am not sure that introducing v_H for just this equation is very useful since it is non-zero in only one entry.
 15. *Below eq. (5)* “The heat absorbed in the thermal and prethermal cycles differs by the work difference at the end of the stroke before the isochore heating, which at the leading order for infinitesimal cycles is half of the total work difference $Q^{th} = Q^{pth} = \frac{1}{2}\delta W$ ”. It is completely unclear how this is true, as the authors did not provide any proof of this.
 16. *Below eq. (5)* Why are the authors only considering the relative efficiency? (they could write it similarly to the previous equations at this point, but they don’t) There are many other typical quantities one could check beyond this. First, how does the efficiency of the prethermal engine compare to Carnot efficiency? Can it beat it since we are out of equilibrium? Is there a similar bound one can find for it? Second, what about power? (And efficiency at maximum power?) Third, what about power fluctuations?
 17. *Results – Prethermal finite cycles in integrable models.* This section seems to be very nicely explained, but it is difficult to understand the presented application of the results of the previous sections because it is not exactly clear what we are looking at.
 18. *Fig. 2* It is very difficult to interpret these plots, and the captions (or text) do not really help the reader to understand what do the graphs tell us. Furthermore the normalization chosen adds a level of encoding that makes it even less understandable. For example, on the green plots: why are we dividing the efficiency by δh ? People are used to efficiency to be bounded by 1, but here you reader does not have any reference. I hope there is a good reason for why one would add the unusual normalization. (also, what are the units of this quantity?)
 19. *Above eq. (8)* “while other choices are associated with different GGEs” for example?
 20. *Fig. 3* Some numbers are being chosen to compute the plots, but no units are provided. I assume some constants have been set to 1 somewhere.

-
21. *Generally on the manuscript.* I usually am against commenting on the presentation style of a manuscript, as it should be a free choice of the authors (within the limits of where they decided to publish their results). However, I believe that here the presentation style has strong negative effects on the communication of the science done by the authors, and is therefore relevant to the discussion. Here the authors chose the quite dry format 1) introduction 2) result A 3) result B ... $N - 1$) conclusion N) methods. I don't think that this is helping in any way the communication of the results. I believe it is actually impairing it, as some things seem very out of context or not introduced at all throughout the manuscript. Very often it is also not clear why the authors study some quantities rather than others. I would like the authors to consider taking a more usual approach where they properly and gradually introduce what is needed to understand their results (including the derivation of the results, as they don't seem very long by looking at the Methods section).